# A neuronal prospect theory model in the brain reward circuitry

Yuri Imaizumi[1], Agnieszka Tymula [2], Yasuhiro Tsubo [3], Masayuki Matsumoto [4] & Hiroshi Yamada [4] ✉

Prospect theory, arguably the most prominent theory of choice, is an obvious candidate for neural valuation models. How the activity of individual neurons, a possible computational unit, obeys prospect theory remains unknown. Here, we show, with theoretical accuracy equivalent to that of human neuroimaging studies, that single-neuron activity in four core reward-related cortical and subcortical regions represents the subjective valuation of risky gambles in monkeys. The activity of individual neurons in monkeys passively viewing a lottery reflects the desirability of probabilistic rewards parameterized as a multiplicative combination of utility and probability weighting functions, as in the prospect theory framework. The diverse patterns of valuation signals were not localized but distributed throughout most parts of the reward circuitry. A network model aggregating these signals reconstructed the risk preferences and subjective probability weighting revealed by the animals' choices. Thus, distributed neural coding explains the computation of subjective valuations under risk.

Since its inception in the 70 s, prospect theory[1] remains one of the most influential descriptive theories of choice in science and social science. The theory proposes that people calculate subjective valuations of risky prospects by a multiplicative combination of two quantities: a value function that captures the subjective value of rewards (i.e., utility) and an inverse S-shaped probability weighting function (i.e., probability weight) that captures a person's subjective distortion of the reward probability when calculating expected utility. The addition of the probability weighting function in their descriptive model of choice under uncertainty allowed Kahneman and Tversky to capture systematic deviations from the expected utility theory, such as Allais Paradoxes[2] and the fourfold pattern of risk attitudes[3]. Prospect theory has been assessed in thousands of studies using behavioral data and is used to explain a plethora of behaviors. However, despite the significant progress in the nascent field of neuroeconomics toward an understanding of how the brain makes economic decisions[4,5], a fundamental question that remains unanswered is whether discharges from individual neurons actually follow the prospect theory model.

Does neuronal activity represent the multiplicative combination of subjective value and probability weighting functions?

Human neuroimaging provides fundamental insights into how economic decision-making is processed by brain activity, especially in the reward circuitry across cortical and subcortical structures[6]. This circuitry is thought to learn the values of rewards and the probability of receiving them through experience[7,8] and it allows human decision-makers to compute subjective valuations of options. Early research in neuroeconomics established that in line with economic theory[9], the brain encodes a utility-like signal that guides choice[10]. At the same time, to establish a biologically viable, unified framework explaining economic decision-making under uncertainty, neuroeconomists aimed to incorporate not only the reward magnitude but also probability into the framework and searched for evidence of inverse-S subjective reward probability weighting in human brain activity using neuroimaging techniques[11–15]. Focusing on the gain domain[13,15], previous studies found that the activity of brain regions in the reward circuitry correlates with individual subjective valuations as proposed

---

[1]Medical Sciences, University of Tsukuba, 1-1-1 Tenno-dai, Tsukuba, Ibaraki 305-8577, Japan. [2]School of Economics, University of Sydney, Sydney 2006 NSW, Australia. [3]College of Information Science and Engineering, Ritsumeikan University, 1-1-1 Noji-Higashi, Kusatsu, Shiga 525-8577, Japan. [4]Division of Biomedical Science, Faculty of Medicine, University of Tsukuba, 1-1-1 Tenno-dai, Tsukuba, Ibaraki 305-8577, Japan. ✉e-mail: h-yamada@md.tsukuba.ac.jp

by the prospect theory[13,15,16]. However, limitations in temporal and spatial resolutions in neuroimaging techniques have restricted our understanding of how the reward circuitry computes subjective valuations of economic decisions, and there have been almost no studies involving the prospect theory analysis of neural mechanisms in the last decade.

Recordings of single-neuron activity in monkeys while receiving risky rewards[17–21] may offer substantial progress over existing neuroimaging studies[11–14]. Specifically, utility coding without probability weighting was tested on the activity of dopamine neurons[22]. Compared to human research, internal valuation measurements of probabilistic rewards have so far been limited to animals, and not all aspects of the prospect theory model could have been measured from animal behavior (e.g.,[23], used only a single probability of 0.5). Recent studies have extended this earlier work by asking whether captive macaques also distort probabilities in the same way humans do[24–27], but no research has yet identified whether the activity of individual neurons in the reward circuitry computes the subjective valuation of risky prospects in a way that is consistent with prospect theory.

Thus, we targeted the reward-related cortical and subcortical structures of non-human primates:[6] the central part of the orbitofrontal cortex (cOFC, area 13 M), the medial part of the orbitofrontal cortex (mOFC, area 14 O), dorsal striatum (DS, the caudate nucleus), and ventral striatum (VS). We measured single-neuron activity in a non-choice situation while monkeys perceived a lottery with a range of probabilities and magnitudes of rewards (10 reward magnitudes by 10 reward probabilities, resulting in 100 unique lotteries). We found many neurons whose activities can be parameterized using the prospect theory model as a multiplicative combination of subjective value (utility) and subjective probability (probability weighting) functions. A simple network model that aggregates these subjective valuation signals, which are distributed through most parts of the reward circuitry, reconstructed the monkey's risk preference and subjective probability weighting estimated from the choices monkeys made in other situations. This is evidence for a neuronal prospect theory model that employs distributed computations in the reward circuitry.

## Results
### Prospect theory and decision characteristics in monkeys
We estimated the monkeys' subjective valuations of risky rewards using a gambling task (Fig. 1a)[28] similar to those used with human

subjects in economics[29]. In the choice trials, monkeys chose between two options that offered a liquid reward with some probability. Monkeys fixated on a central gray target, and then two options were presented visually as pie charts displayed on the left and right sides of the screen. The number of green pie segments indicated the magnitude of the liquid reward in 0.1 mL increments (0.1–1.0 mL), and the number of blue pie segments indicated the probability of receiving the reward in 0.1 increments (0.1–1.0, where 1.0 indicates a 100% chance). Monkeys chose between left and right targets by fixating on one side. Following the choice, monkeys received or did not receive the amount of liquid reward associated with their chosen option, according to their corresponding probability. In each choice trial, two out of 100 possible combinations of the probability and magnitude of rewards were randomly selected and allocated to the left- and right-side target options. We used all the data collected after each monkey learned to associate the probability and magnitude with the pie-chart stimuli. This included 44,883 decisions made by monkey SUN (obtained in 884 blocks over 242 days) and 19,292 decisions made by monkey FU (obtained in 571 blocks over 127 days). These well-trained monkeys, like humans, showed behavior consistent with utility maximization, selecting average options with a higher expected value, that is, probability times magnitude (Fig. 1b). In the experiment, a block of choice trials was occasionally interleaved with a block of single-cue trials (Fig. 1c) during which neural recordings were made. In these trials, monkeys did not make a choice but passively viewed a single lottery cue, which offered some reward with some probability given after a delay.

We estimated each monkey's utility and probability weighting functions from its choice behavior using standard parametrizations in the literature. For the utility function, we used the power utility function $u(m) = m^\alpha$, where $m$ indicates the magnitude of the reward, $\alpha > 1$ indicates convex utility (risk-seeking behavior), $\alpha < 1$ indicates concave utility (risk aversion), and $\alpha = 1$ indicates linear utility (risk neutrality). For the probability weighting function $w(p)$, we used one-parameter, $w(p) = exp(- (-log p)^\gamma)$, and two-parameter, $w(p) = exp(-\delta (-log p)^\gamma)$, Prelec functions. The one-parameter version was nested in the two-parameter version (when $\delta = 1$) for ease of comparison. Overall, we estimated the following four models of the utility of receiving a reward magnitude $m$ with probability $p$, $V(p,m)$:

1. EV: expected value

$$V(p, m) = p\,m \tag{1}$$

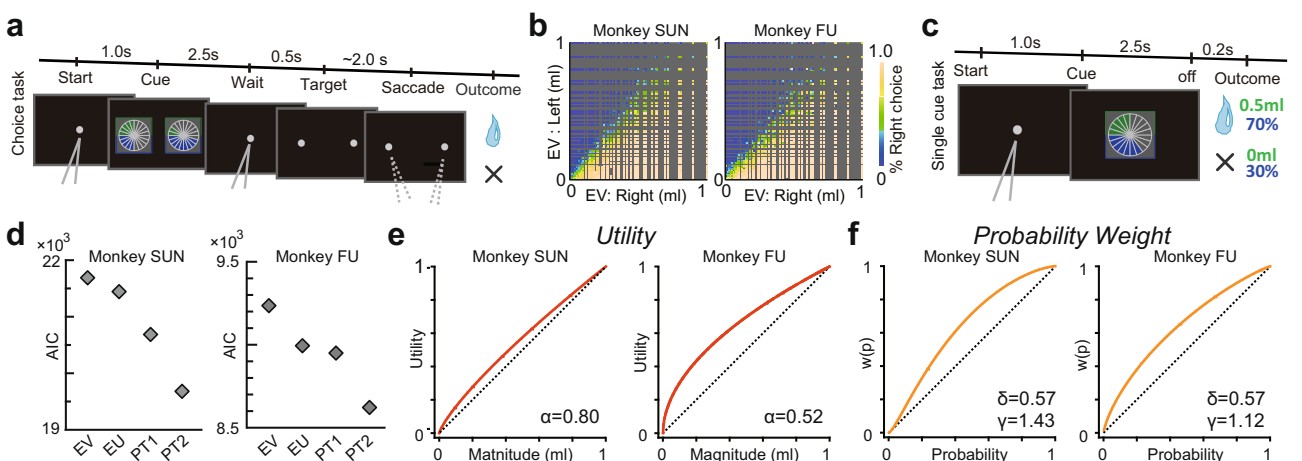

**Fig. 1 | Cued lottery task and monkeys' choice behavior. a** A Sequence of events in the choice trials. Two pie charts representing available options were presented to the monkeys on the left and right sides of the screen. The monkeys chose either of the targets by fixating on the side where they appeared. **b** Frequency with which the target on the right side was selected for the expected values of the left and right target options. **c** Sequence of events in single-cue trials. **d** AIC values are estimated based on the four standard economic models to describe the monkey's choice behavior: EV, EU, PT1, and PT2. See the Methods section for details. **e** Estimated utility functions in the best-fit model PT2. **f** Estimated probability-weighting functions in the best-fit model PT2. Images in panels a-c were created by the authors and previously published in Neural Population Dynamics Underlying Expected Value Computation. Hiroshi Yamada, et al.[28]. https://creativecommons.org/licenses/by/4.0/.

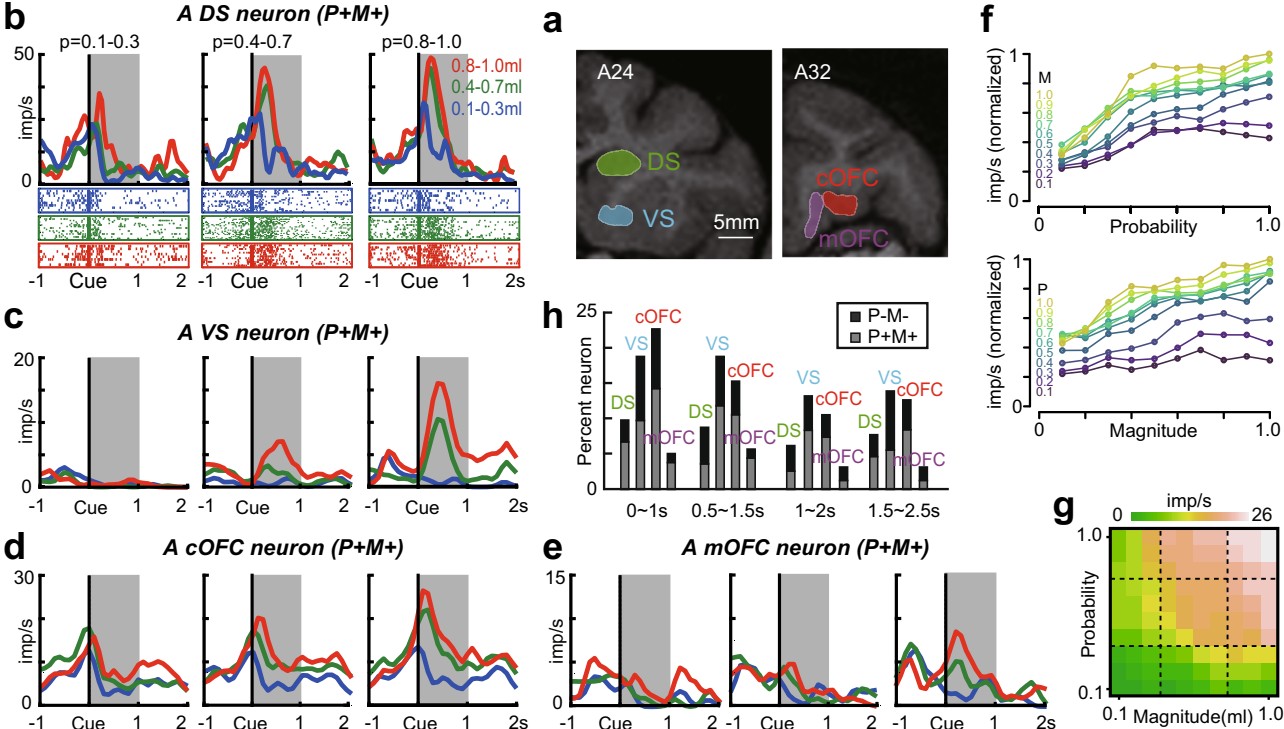

**Fig. 2 | Neural coding of probability and magnitude of rewards in the four brain regions. a** Illustration of neural recording areas based on coronal magnetic resonance images. **b** Example activity histogram of a DS neuron modulated by probability and magnitude of rewards with positive regression coefficients during the single-cue task ($P + M +$ type). The activity aligned to the cue onset is represented for three different levels of probability (0.1–0.3, 0.4–0.7, and 0.8–1.0) and magnitude (0.1–0.3 mL, 0.4–0.7 mL, and 0.8–1.0 mL) of rewards. Gray hatched time windows indicate the 1-s time window used to estimate the neural firing rates shown in **f** and **g**. Raster grams are shown below. **c–e** similar to **b**, but for VS, cOFC, and mOFC neurons. **f** Plot of the neural firing rates during the 1-s time window in **b** for ten levels of probability and magnitude of rewards. The firings are normalized by the maximum firing rates. P and M indicate the probability and magnitude of rewards, respectively. **g** Color map of the neural firing rates during the 1 s time window in **b** for ten levels of probability and magnitude of rewards. Average smoothing was made between neighboring pixels. **h** Percentage of neurons modulated by probability and magnitude of rewards in the four core reward brain regions. Gray indicates activity showing positive regression coefficients for probability and magnitude of rewards ($P + M +$ type). Black indicates activity showing the negative regression coefficients for probability and magnitude (P-M- type). Images in panels a were created by the authors and previously published in Neural Population Dynamics Underlying Expected Value Computation. Hiroshi Yamada, et al.[28]. https://creativecommons.org/licenses/by/4.0/.

2. EU: expected utility

$$V(p, m) = p\, m^{\alpha} \quad (2)$$

3. PT1, one-parameter Prelec: prospect theory with $w(p)$ as in[30]

$$V(p, m) = exp(-(-log\, p)^{\gamma})\, m^{\alpha} \quad (3)$$

4. PT2, two-parameter Prelec: prospect theory with $w(p)$ as in[31]

$$V(p,m) = exp(-\delta(-log\, p)^{\gamma})\, m^{\alpha} \quad (4)$$

$\alpha$, $\delta$, and $\gamma$ are free parameters and $p$ and $m$ are the probability and magnitude of the reward cued by the lottery, respectively. The parameters $\delta$ and $\gamma$ control the subproportionality and regressiveness of $w(p)$. We assumed that subjective probabilities and utilities were integrated multiplicatively, as is customary in economic theory, yielding $V(p, m) = w(p)\, u(m)$. The probability of the monkey choosing the lottery on the right side ($L_R$) instead of the lottery on the left side ($L_L$) was estimated using the logistic choice function:

$$P(L_R) = 1/(1 + e^{-z}) \quad (5)$$

where $z = \beta (V(L_R)\ V(L_L))$ and the free parameter $\beta$ controls the degree of stochasticity observed in the choices.

To determine which model best describes the behavior of a monkey, we used Akaike's information criterion (AIC), which measures the goodness of model fit with a penalty for the number of free parameters employed by the model (see Methods section for more details). Among the four models, PT2 had the lowest AIC and outperformed EV, EU, and PT1 in both monkeys (Fig. 1d, 44,883 and 19,292 trials in monkey SUN and monkey FU, respectively). In the best-fit model, the utility function was concave (Fig. 1e; one-sample t-test: df = 44,882, $\alpha = 0.80$, z = 46.10, $P < 0.001$ in monkey SUN; df = 19,291, $\alpha = 0.52$, z = 25.04, $P < 0.001$ in monkey FU), indicating that the monkeys were risk-averse. Notably, for both monkeys, the probability weighting functions were concave instead of the inverse-S shape traditionally assumed in humans (Fig. 1f; one-sample t-test: df = 44,882, $\delta = 0.57$, z = 86.51, $P < 0.001$ in monkey SUN; $\delta = 0.57$, z = 52.77, $P < 0.001$ in monkey FU; df = 19,291, $\gamma = 1.43$, z = 47.29, $P < 0.001$ in monkey SUN; $\gamma = 1.12$, z = 25.68 in monkey FU, $P < 0.001$). Overall, we conclude that in monkeys, utility functions estimated from behavior are concave, similar to those in humans, but monkeys distort probability differently compared to what is usually assumed by human decision-makers.

### Subjective valuation signals were distributed in the reward circuitry

We recorded single-neuron activity during the single-cue task (Fig. 1c) from neurons in the DS (n = 194: monkey SUN, 98; monkey FU, 96), VS (n = 144: monkey SUN, 89; monkey FU, 55), cOFC (n = 190: monkey SUN, 98; monkey FU, 92), and mOFC (n = 158: monkey SUN, 64; monkey FU, 94) (Fig. 2a). These brain regions are known to be involved in decision-making. We first identified neurons whose activity represents

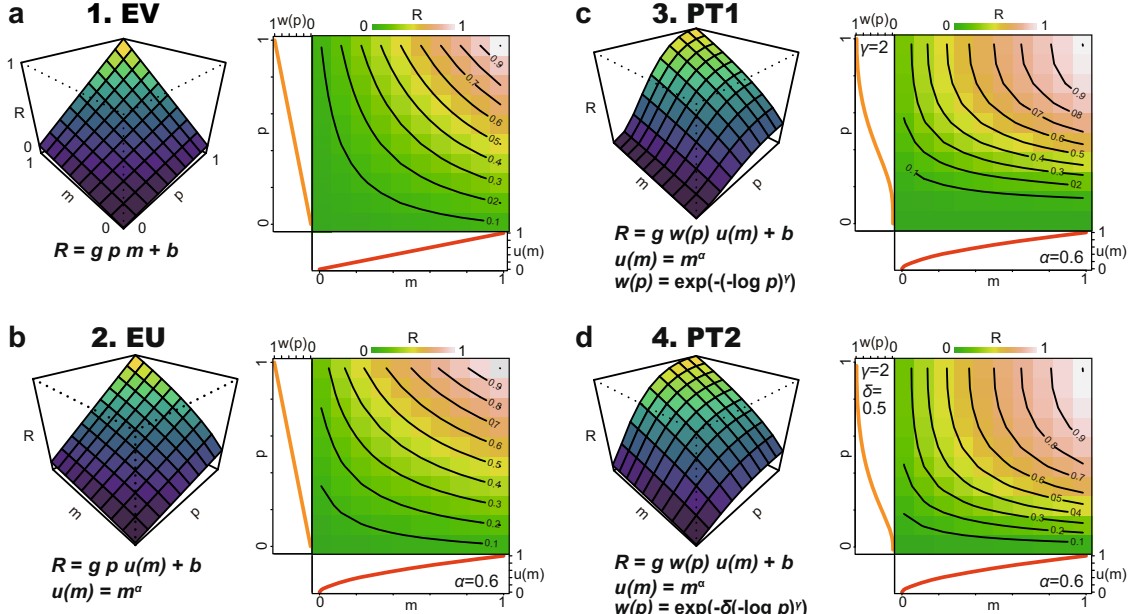

**Fig. 3 | Neural models of economic decision theory.** Schematic depiction of predicted neuronal responses *R* defined by the four economic models that represent the expected value (**a**, EV), expected utility (**b**, EU), prospect theory one-parameter Prelec (**c**, PT1), and two-parameter Prelec (**d**, PT2). Model equations are presented in each plot. *R* was plotted against the probability (*p*) and magnitude (*m*) of the rewards. *b*, *g*, *α*, *γ*, and *δ* are the free parameters. *g* and *b* are the gain and intercept parameters, respectively. *α* represents the curvature of *u*(*m*). *δ* and *γ* represent the probability weighting functions. For these schematic drawings, the following values for the free parameters were used: *b*, *g*, *α*, *γ*, and *δ* were 0 spk s$^{-1}$, 1, 0.6, 2, and 0.5, respectively, for all four Figs. See the Methods section for more details.

the key reward statistics—probability and magnitude—that underlie the expected value, expected utility, and prospect theory. Because after the presentation of lottery cue, neurons in the four brain regions showed a firing rate increase with a variable time course (Fig. 1h in Yamada et al., 2020), we analyzed the neural firing rate through the cue period with four analysis epochs (see Methods). These neurons were identified by regressing neural activity on probability and magnitude of rewards, and the neurons included in our analysis were those that had either both positive or both negative regression coefficients (see Methods), which is the potential signature of *V*(*p,m*) - the decision statistics in economic theory.

An example of single neuron activity during a 1 s time window after cue onset is shown in Fig. 2b. This DS neuron showed activity modulated by both the probability and magnitude of rewards with positive regression coefficients (P + M + type, proportion of variance explained, 0.462, *n* = 114; probability, regression coefficient, β = 13.51, *t* = 8.57, *P* < 0.001; magnitude, β = 12.27, *t* = 7.79, *P* < 0.001). These types of neurons were also observed in the VS, cOFC, and mOFC (Fig. 2c–e, VS, P + M + type, proportion of variance explained, 0.440, *n* = 115; probability, regression coefficient, β = 7.14, *t* = 7.31, *P* < 0.001; magnitude, β = 6.71, *t* = 6.81, *P* < 0.001; cOFC, P + M + type, proportion of variance explained, 0.509, *n* = 119; probability, regression coefficient, β = 8.55, *t* = 6.91, *P* < 0.001; magnitude, β = 11.07, *t* = 8.95, *P* < 0.001; mOFC, P + M + type, proportion of variance explained, 0.238, *n* = 120; probability, regression coefficient, β = 2.72, *t* = 3.95, *P* < 0.001; magnitude, β = 2.88, *t* = 4.15, *P* < 0.001). Neuronal firing rates increased as the reward probability increased and as the reward magnitude increased, representing a positive coding type (Fig. 2f). In a plot of neuronal activity for all combinations of probability and magnitude, a curvature of the neural firing rates was detected (Fig. 2g). Similarly, some neurons showed activity modulated by both the probability and magnitude of rewards with negative regression coefficients, representing a negative coding type (P-M- type, Supplementary Fig. 1). In total, these types of activity were observed in 24% (164/686) of all recorded neurons in at least one of the four analysis epochs

during the 2.5 s cue period. The proportions of these signals in each brain region were different (DS, 22%, 43/194, VS, 32%, 45/141, cOFC, 31%, 59/190, mOFC, 11%, 17/158, chi-square test, X$^2$ = 25.59, df = 3, *P* < 0.001) with significant differences in DS and mOFC between monkeys (DS: monkey SUN 32/98, monkey FU 11/96, X$^2$ = 11.43, df = 1, *P* < 0.001; VS: monkey SUN 30/89, monkey FU 15/52, X$^2$ = 0.17, df = 1, *P* = 0.682; cOFC: monkey SUN 33/98, monkey FU 26/92, X$^2$ = 0.42, df = 1, *P* = 0.52; mOFC: monkey SUN 15/64, monkey FU 2/94, X$^2$ = 16.26, df = 1, *P* < 0.001). These neurons were evident across the entire cue period (Fig. 2h), during which the monkeys perceived the probability and magnitude of rewards. Thus, cue period activity in the four core reward brain regions showed potential signature of *V*(*p, m*), which is the core decision statistics in economic theory.

We also found that the activity of neurons modulated by either probability or magnitude (probability, 305/686 neurons; magnitude, 269/686 neurons; at least one of the four analysis epochs) and across the entire cue period (probability: 0-1 s 108/686, 0.5-1.5 s 133/686, 1-2 s 128/686, 1.5-2.5 s 146/686; magnitude: 0-1 s 115/686, 0.5-1.5 s 113/686, 1-2 s 108/686, 1.5-2.5 s 113/686). We did not further analyze this activity of neurons because our main focus was on the *V*(*p, m*).

### Detecting the neuronal signature of prospect theory

To visually inspect the potential neuronal signature of *V*(*p,m*), we predicted from the behavioral estimates how the observed neuronal firing rates should look in each of the four models: expected value (Fig. 3a, EV), expected utility (Fig. 3b, EU), and prospect theory (Fig. 3c and d, PT1 and PT2, respectively). In each of the models, the neural firing rate *R* is given by

$$R = g\,w(p)\,u(m) + b \tag{6}$$

where the predicted neuronal response *R*, the output of the model, integrates the subjective value function (i.e., utility, *u*(*m*)) and subjective probability function (i.e., probability weight, *w*(*p*)). *b* is a free parameter that captures the baseline firing rates in the probability-

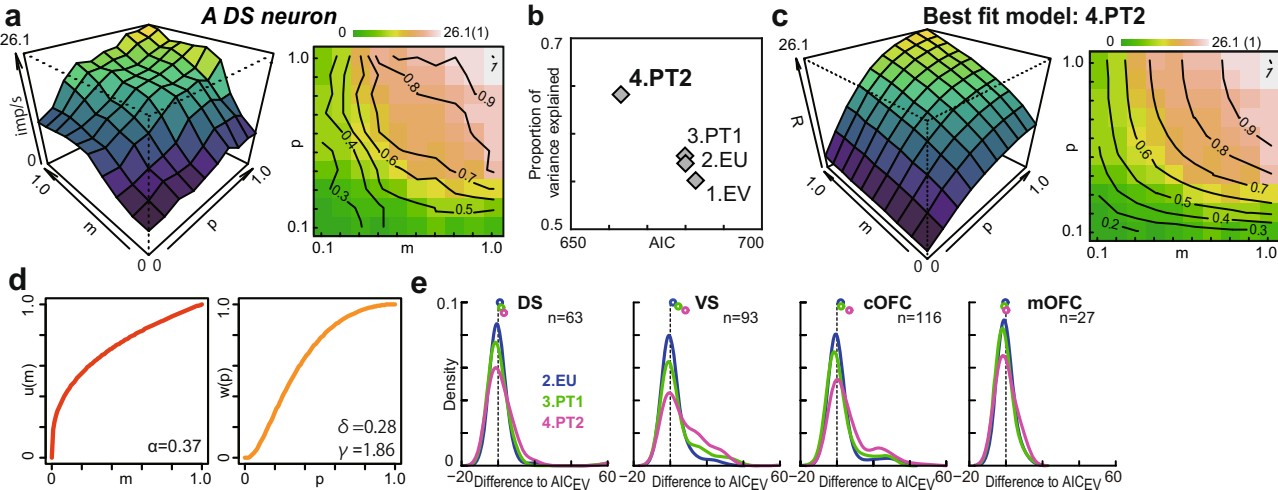

**Fig. 4 | Prospect theory best explained neural firing rates in the reward circuitry. a** Plot of an example activity of the DS neuron in Fig. 2b against the probability (*p*) and magnitude (*m*) of rewards. To draw the 3D curvature (left) and contour lines (right), the neighboring pixels were averaged and smoothed. **b** AIC values against the proportion of variance explained are plotted in each model for the example neuron in **a**. **c** A 3D histogram (left) and contour lines (right) predicted from the best-fit PT2 model in **a**. The activity of the example neuron in **a** is shown on the right color map. Contour lines are shown for every 10% change in the fit model. **d** *u(m)* and *w(p)* estimated in the best-fit model PT2 for the neural activity in **a**. **e** Probability density of the estimated AIC difference of the three models against the EV (simplest) model. The plots display the mean values. n represents the number of neuronal signals that showed both positive and negative regression coefficients for the probability and magnitude of the rewards.

magnitude space. *g* determines how strongly the magnitude of the neural response depends on *u(m)* and *w(p)*. *u(m)* and *w(p)* are specified for each model, as described above (see formulas in Fig. 3 and Methods section).

Next, we aimed to assess which of the models best captured neuronal discharge rates in each brain region. Therefore, we fitted the activities of individual neurons with each of the four models, treating *b*, *g*, *α*, *δ*, and *γ* as the free parameters. Our carefully designed set of lottery stimuli—a sampling matrix of 10 rewards by 10 probabilities— allowed us to perform a reliable estimation of these five free parameters for each neuronal activity. To determine which model best described the observed neuronal firing rate in each individual neuron, we used the AIC. As demonstrated for the example neuron in Fig. 4a, the activity of this DS neuron was best explained by prospect theory with a two-parameter probability weighting function (Fig. 4b, PT2). For this neuron, PT2 had the smallest AIC value with the highest proportion of explained variance. The output *R* of the fitted PT2 model described the activity pattern well (Fig. 4c) as well as the observed activity (Fig. 4a), in which the neural utility function and subjective probability weighting function were parameterized (Fig. 4d) via a multiplicative relation in the model.

To understand which model best describes the neural activity in each brain region, we determined the goodness-of-fit score for each neuronal activity as the difference in AIC between each of the models (EU, PT1, and PT2) and the EV model. Here, we treated the EV model as the baseline because it is the simplest model and a predecessor of other models in the economics literature. Figure 4e shows the probability density of the goodness-of-fit score differences for each brain region separately. The vertical dashed lines at 0 indicate no difference between the AIC of the EV model and that of the model under consideration. The model that shows a greater deviation to the right of the graph indicates a better fit.

Overall, prospect theory (PT2) best described the activity of most neural populations in the reward circuitry (DS, VS, and cOFC), except for mOFC activity. The AIC values of the four models were statistically compared. Comparisons indicated that the PT2 model was best at describing DS, VS, and cOFC activity as a whole (one-sample *t*-test after subtracting the models' AIC scores; DS: *n* = 63, df = 62, EV-EU, *t* = 0.94,

*P* = 0.35, EU-PT1, *t* = 1.03, *P* = 0.31, PT1-PT2, *t* = 3.01, *P* = 0.004; VS: *n* = 93, df = 92, EV-EU, *t* = 2.42, *P* = 0.017, EU-PT1, *t* = 4.00, *P* < 0.001, PT1-PT2, *t* = 3.91, *P* < 0.001; cOFC: *n* = 116, df = 115, EV-EU, *t* = 2.90, *P* = 0.004, EU-PT1, *t* = 0.65, *P* = 0.52, PT1-PT2, *t* = 6.18, *P* < 0.001, not shown for all). However, the best descriptive model of the mOFC activity could not be determined (one-sample *t*-test; mOFC: *n* = 27, df = 26, EV-EU, *P* = 0.60, EU-PT1, *P* = 0.10, PT1-PT2, *P* = 0.11), suggesting that mOFC activity simply signals expected values without distorting the objective probability and magnitude of rewards during the perception of the lottery.

Next, we asked whether some neuronal populations specifically encoded subjective valuations based on their location (DS, VS, and cOFC). For this purpose, we used the PT2 model estimates, *b*, *g*, *α*, *δ*, and *γ* of the individual activity of neurons, including both positive and negative coding types. We clustered these five parameters using k-means clustering algorithms following principal component analysis (PCA) across the neural populations in the DS, VS, and cOFC (Figs. 5a and 5b, see Methods). Five predominant clusters, C1 to C5, were obtained after PCA based on the four principal components (Fig. 5b). These five clusters were observed in similar proportions across the three brain regions with only slight differences (Fig. 5c). One small difference was that VS contained a smaller proportion of the predominant cluster, C1, than the other two regions (chi-square test, *n* = 272, X² = 18.15, df = 8, *P* = 0.020).

Across the DS, VS, and cOFC, C1 represented 48% of all activities (Fig. 5d, top row; *n* = 130, mean values: *b* = −0.68, *g* = 10.1, *α* = 0.64, *δ* = 1.30, *γ* = 2.64). Its output, *R*, is described by a combination of a concave utility function and an S-shaped probability weighting function (Fig. 5d, see the third and fourth columns in the top row). The second predominant cluster, C2, was best described with a concave utility function, but its probability weighting function was concave. This cluster was mostly composed of neurons with negative coding of the probability and magnitude of rewards (Fig. 5d, middle row; *n* = 78, *b* = 10.6, *g* = −10.1, *α* = 0.29, *δ* = 0.38, *γ* = 1.82). Because the coding gain was negative (Fig. 5d, middle left, note that the axis values are plotted from 1.0 to 0), the convex curvature (Fig. 5d, left column in the middle row) of the output *R* corresponded to the concave functions *u(m)* and *p(w)*. A considerable proportion of neurons (9%), C3, showed an output

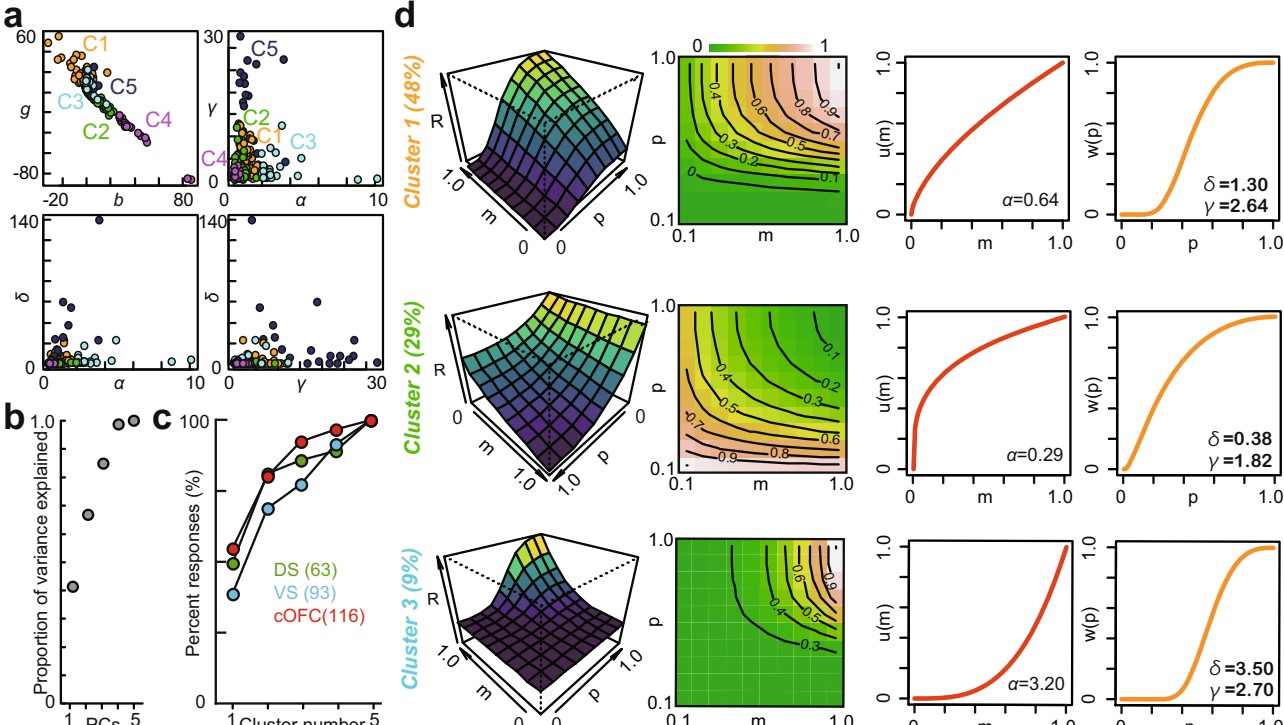

**Fig. 5 | Neuronal clusters categorized by the fitted parameters according to the prospect theory model. a** Plots of all five parameters estimated in DS, VS, and cOFC neurons. $g$, $b$, $\alpha$, $\delta$, and $\gamma$ were plotted. **b** Cumulative plot of the proportion of variance explained by PCA is shown against principal components PC1–PC5. **c** Cumulative plot of the percentage of activity categorized into five clusters in each brain region. **d** Response $R$ (model output) in the first three predominant clusters is plotted. The 3D curvature, contour lines with color maps, $u(m)$, and $w(p)$ are plotted using the mean values of each parameter in each cluster. To draw the 3D curvature (first column) and contour lines (second column), $R$ was normalized to the maximal value.

well described by a convex utility function and an S-shaped probability weighting function (Fig. 5d, bottom; $n = 25$, $b = 2.6$, $g = 7.2$, $\alpha = 3.2$, $\delta = 3.5$, $\gamma = 2.7$). These clusters of neuronal activities parameterized by the prospect theory model were not localized and were instead scattered across most parts of the reward circuitry (DS, VS, and cOFC), suggesting that distributed coding underlies internal subjective valuations under risk.

### Reconstruction of internal preference parameters from observed neural activity

Finally, we reconstructed the monkeys' internal valuations of passively viewed lotteries from the observed neural activity to assess how well they matched the utility and probability weighting functions estimated from the behavioral choices. To do so, we constructed a simple three-layered network model as a minimal rate model, a primitive version of the advanced models used recently[32,33], and simulated the choices of this network model (Fig. 6). We assumed that outputs reflecting $V(p, m)$ in neural clusters C1 to C5 (Fig. 6a, Rs in the first layer) were linearly integrated and positively rectified by the network (Fig. 6a, second layer, population SEV, see Methods). The activities in clusters 1, 3, and 5 (mostly composed of P + M + neurons) were linearly summed, and those in clusters 2 and 4 (mostly composed of P-M- neurons) were subtracted to integrate the opposed signals (hence, linear summation of an inversed signal). To simulate the choice, we generated two identical population SEVs for the left ($SEV_L$) and right ($SEV_R$) target options and used a random utility model to select one option (Fig. 6a, third layer, sigmoid choice function). Overall, we simulated 40,000 choices–four times for each possible combination of 100 lotteries, $L(p, m)$.

While our network model used neural signals modeled by prospect theory during passive viewing, these simulated choice patterns based on the clustered neuronal prospect theory model were very

similar to the actual gambling behaviors of monkeys (Figs. 6b and 1b). When estimating the utility function and probability weighting function of these simulated choices, we observed concave utility functions and concave probability weighting functions similar to those obtained from actual gambling behavior (Fig. 6c). We repeated this simulation 1000 times to construct the parameter distributions of the internal subjective valuation obeyed by the layered model. The mean and standard deviation of the estimated parameters were as follows: alpha, $0.49 \pm 0.017$; delta, $0.50 \pm 0.018$; gamma, $1.67 \pm 0.014$. They were significantly different from 1 ($P < 0.001$ for all cases). Thus, we concluded that a distributed neural code that accumulates individual neuronal signals explains the internal subjective valuations of monkeys.

## Discussion

The prospect theory is the dominant theory of choice in behavioral economics, but it remains elusive whether the theory is only descriptive of human behavior or has a deeper meaning in the sense that it also describes an underlying neuronal computation that extends to our close evolutionary relatives. Previous human neuroimaging studies have demonstrated that neural responses to rewards measured through blood oxygen levels can be described using prospect theory[13,15,16] but with limited resolution in the temporal and spatial domains. Here, we provide the first evidence that the activity of individual neurons in the reward circuitry (DS, VS, and cOFC) of monkeys that perceive a lottery can be captured based on the prospect theory model as a multiplicative combination of utility and probability weighting functions (Fig. 4). This is consistent with the idea suggested by Tobler et al. (2007)[34] that the striatum integrates the reward magnitude and probability via multiplication into an expected value signal. Previous human fMRI studies found a nonlinear response to probability in striatal regions[12,15] and the

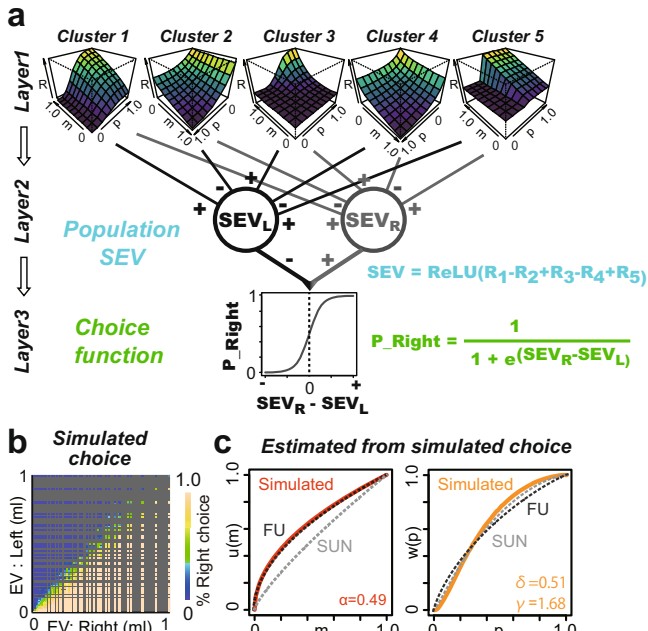

**a**

Cluster 1   Cluster 2   Cluster 3   Cluster 4   Cluster 5

Layer1

Population SEV

$SEV = ReLU(R_1-R_2+R_3-R_4+R_5)$

Layer2

Choice function

$P\_Right = \dfrac{1}{1 + e^{(SEV_R - SEV_L)}}$

Layer3

**b** *Simulated choice*

**c** *Estimated from simulated choice*

Simulated — FU / SUN — $\alpha=0.49$

Simulated — FU / SUN — $\delta=0.51$, $\gamma=1.68$

**Fig. 6 | A simple network model reconstructs the subjective decision statistics in monkeys. a** Five neural clusters detected by PCA in the reward circuitry. To draw the 3D curvature, $R$ was normalized to the maximal value. The subjective expected value functions (SEV) for the left and right target options are defined as the linear summation of the five clusters (see Methods). The choice was simulated as a sigmoid function of the SEV's signal difference. **b** Frequency with which the target on the right side was selected by a computer simulation based on the network shown in **a**. **c** $u(m)$ and $w(p)$ estimated from the simulated choice data in **b** are plotted. The dotted lines indicate the actual functions $u(m)$ and $w(p)$ of the monkeys, as shown in Fig. 1e and f, respectively.

dorsolateral prefrontal cortex[34], while some studies concluded that the probability coding in the striatum is linear[11,13,14]. The probability weighting that we estimated in behavior and recovered from neural activity is concave.

One pivotal question is how various subjective preference signals are transformed into behavioral choices through information processing via neural networks. Our clustering analysis of parameterized neuronal activity revealed that these signals were similarly distributed across the VS, DS, and cOFC (Fig. 5). Our minimal rate model of a three-layered network successfully reconstructed the internal valuation of risky rewards observed in monkeys (Fig. 6), suggesting that these subjective valuation signals in the reward circuitry would be integrated into the brain to construct a decision output from risky perspectives. It is important to denote that influences of $V(p,m)$ signals on the simulation might be different between monkeys, since larger number of the detected $V(p,m)$ signals in monkey SUN must affect the clustering of $V(p,m)$ signals.

Previous studies have shown that neuronal signals related to cognitive and motor functions are widely observed in many brain regions[35–41]. These distributed neuronal signals suggest that the distributed neural code is a canonical computation in the brain. The recent development of large-scale neural recording technologies has verified that this is a common computational mode;[42] the analysis of approximately 30,000 neurons in 42 regions of the rodent brain revealed that behaviorally relevant task parameters are observed throughout the brain. Our results from the reward-related brain regions are in line with this view, except for the mOFC, where fewer encodings of probability and magnitude of rewards were observed (Fig. 2h), with a significant difference between monkeys. This might be because the medial-lateral axis in the reward circuitry yields a significant difference in reward-based decision-making[6]. The distributed

code may require some input-output functions[43] to process the probability and magnitude of rewards and integrate this information to estimate the subjective expected value signals, at least in some neural populations. One possible information processing for this input-output mapping might be achieved by neural population dynamics[44–46], in which some subclusters of neurons can process information moment-by-moment as a dynamical structure of information processing in a neural network. Indeed, stable neural population dynamics in the VS and cOFC were observed in contrast to the fluctuating signals in the DS population[28] with a significant difference between monkeys in the present study, which may reflect some dynamic differences in distributed coding in each individual.

One limitation of our study is that our application of prospect theory is limited to the domain of gains, since unlike human studies that use money as a reward, it is impossible to take fluid rewards from monkeys to make them experience losses. Nevertheless, our study adds important behavioral evidence to the growing literature on prospect theory preference in primates. Recent studies on captive macaques have begun to investigate the possibility that monkeys make decisions based on probability values different from those that are objectively correct, with inconsistent results across studies[24–27,47,48]. The probability weighting function was inverse S-shaped[25,26], S-shaped[24,27], or concave[26,49]. Although we consistently found that the probability weighting functions of our two well-trained monkeys were concave, most studies conducted in humans found inverse-S-shaped probability weighting functions at the aggregate level, with a large amount of heterogeneity at the individual level[13,15,50–54] indicating an inconsistency between the two species. Furthermore, the monkeys in the present study had concave utility functions (i.e., risk aversion), while most previous studies have found that monkeys have convex (i.e., risk-seeking)[24,25] or concave[23,47–49] utility over rewards in the gain domain. In conclusion, our monkeys had concave utility functions, similar to our previous findings in monkeys[23,55] as well as in humans. However, unlike humans, our monkeys have concave probability-weighting functions.

In summary, we provide novel evidence that the activity of individual neurons in the reward circuitry can be described using prospect theory. These aggregated signals reliably reconstructed the risk preferences and subjective probability weighting estimated from the monkeys' behavior. We note that the probability weighting in our study is different from that assumed by Kahneman and Tversky (1979) assumed for humans[1].

## Methods

### Subjects and experimental procedures
Two rhesus monkeys were used (*Macaca mulatta*, SUN, 7.1 kg, male, during 4-8 years old; *Macaca fuscata*, FU, 6.7 kg, female, during 4–7 years old). All experimental procedures were approved by the Animal Care and Use Committee of the University of Tsukuba (Protocol No. H30.336) and performed in compliance with the US Public Health Service's Guide for the Care and Use of Laboratory Animals. Each animal was implanted with a head-restraint prosthesis. Eye movements were recorded at 120 Hz using a video camera. Visual stimuli were generated using a liquid-crystal display at 60 Hz, placed 38 cm from the monkey's face when seated. Our recording system used Matlab R2015b with Psychtoolbox 3.0 for behavioral task control. Open developer software 2.16, OpenEx 2.16, and OpenSorter 2.16 were used in TDT system for data collection. The subjects performed the cued lottery task 5 days a week. The subjects practiced the cued lottery task for 10 months, after which they became proficient in choosing lottery options.

### Cued lottery tasks
Animals performed one of two visually cued lottery tasks: a single-cue task or a choice task.

**Single-cue task**. At the beginning of each trial, monkeys had 2 s to align their gaze within 3° to a 1° diameter gray central fixation target. After fixation for 1 s, an 8° pie chart providing information on the probability and magnitude of rewards was presented for 2.5 s at the same location as the central fixation target. The probability and magnitude were indicated by the number of blue and green pie chart segments, respectively. The pie chart was then removed and 0.2 s later, a 1 kHz and 0.1- kHz tone of 0.15 s duration indicated the reward and no-reward outcomes, respectively. The high tone preceded reward delivery by 0.2 s, whereas the low tone indicated that no reward was delivered. The animals received a liquid reward, as indicated by the number of green pie chart segments, with the probability indicated by the number of blue pie chart segments. An intertrial interval of 4–6 s was used for each trial.

**Choice task**. At the beginning of each trial, monkeys had 2 s to align their gaze within 3° to a 1° diameter gray central fixation target. After fixation for 1 s, two peripheral 8° pie charts providing information on the probability and magnitude of rewards for each of the two target options were presented for 2.5 s at 8° to the left and right of the central fixation location. The gray 1°-choice targets appeared at the same location. After a 0.5 s delay, the fixation target disappeared, resulting in saccade initiation. Monkeys were allowed 2 s to make their choice by shifting their gaze to either target within 3° of the chosen target. A 1 kHz and 0.1 kHz tone sounded for 0.15 s to denote reward and no-reward outcomes, respectively. The animals received a liquid reward, as indicated by the number of green pie chart segments of the chosen target, with the probability indicated by the number of blue pie chart segments. An intertrial interval of 4–6 s was used for each trial.

**Payoff, block structure, and data collection**. Green and blue pie charts respectively indicated reward magnitudes from 0.1 to 1.0 mL, in 0.1 mL increments, and reward probabilities from 0.1 to 1.0, in 0.1 increments. A total of 100 pie chart combinations were used in this study. In the single-cue task, each pie chart was presented once in random order, allowing monkeys to experience all 100 lotteries within a certain period. In the choice task, two pie charts were randomly allocated to the left and right targets for each trial. Approximately 30–60 trial blocks of the choice task were sometimes interleaved with the 100–120 trial blocks of the single-cue task.

**Calibration of the reward supply system**. A precise amount of the liquid reward was delivered to monkeys using a solenoid valve. An 18-gauge tube (0.9 mm inner diameter) was attached to the tip of the delivery tube to reduce variation across trials. The amount of reward in each payoff condition was calibrated by measuring the weight of water with 0.002 g precision (2 μL) on a single trial basis. This calibration method was the same as that used in[55].

**Electrophysiological recordings**. Conventional techniques were used to record single-neuron activity in the DS, VS, cOFC, and mOFC. Monkeys were implanted with recording chambers (28 × 32 mm) targeting the OFC and striatum, centered 28 mm anterior to stereotaxic coordinates. The locations of the chambers were verified using anatomical magnetic resonance imaging. A tungsten microelectrode (1–3 MΩ, FHC) was used to record neurons. The electrophysiological signals were amplified, band-pass filtered, and monitored. Single-neuron activity was isolated based on spike waveforms. We recorded from the four brain regions of a single hemisphere of each of the two monkeys: 194 DS neurons (98 and 96 from monkeys SUN and FU, respectively), 144 VS neurons (89 from SUN and 55 from FU), 190 cOFC neurons (98 from SUN and 92 from FU), and 158 mOFC neurons (64 from SUN and 94 from FU). The activity of all the single neurons was sampled when the activity of an isolated neuron demonstrated a good signal-to-noise ratio (>2.5). Blinding was not performed. The sample

sizes required to detect effect sizes (the number of recorded neurons, the number of recorded trials in a single neuron, and the number of monkeys) were estimated in reference to the previous studies[44,55,56]. Neural activity was recorded during the 100–120 trials of the single-cue task. During the choice trials, neural activity was not recorded. Presumed projection neurons (phasically active neurons[57],) were recorded from the DS and VS, whereas presumed cholinergic interneurons (tonically active neurons[58,59],) were not recorded.

## Statistical analysis
Statistical analysis was performed using statistical software R and Stata. All statistical tests were two-tailed. We used standard maximum likelihood procedures to estimate the utility functions and probability weighting functions in Stata. We performed neural analysis and simulation to reconstruct the choice of a neural model in R.

## Behavioral analysis
We first examined whether the choice behavior of monkeys depended on the expected values of the two options located on the left and right sides of the screen. We pooled choice data across all recording blocks (monkey SUN: 884 blocks, 242 days; monkey FU: 571 blocks, 127 days), yielding 44,883 and 19,292 choice trials for monkeys SUN and FU, respectively. The percentage of right target choices was estimated from the pooled choice data for all combinations of the expected values of the left and right target options. This result has been reported previously[28].

## Economic models
We estimated the parameters of the utility and probability weighting functions within a random utility framework. Specifically, lottery $L(p,m)$ denoted a gamble that paid $m$ (magnitude of the offered reward in mL) with a probability $p$ or 0 otherwise. We assumed a popular constant relative risk attitude (also known as the power utility function), $u(m) = m^{\alpha}$, and considered previously proposed probability weighting functions. We assumed two subjective probability weighting functions $w(p)$ commonly used in the prospect theory; one-parameter Prelec (PT1): $w(p) = exp(- (-log\ p)^{\gamma})$[30] and two-parameter Prelec (PT2): $w(p) = exp(-\delta\ (-log\ p)^{\gamma})$[31]. We assumed that subjective probabilities and utilities were integrated multiplicatively per standard economic theory, yielding the expected subjective utility function $V(p,m) = w(p)\ u(m)$.

The probability of a monkey choosing the lottery on the right side ($L_R$) instead of the lottery on the left side ($L_L$) was estimated using the logistic choice function:

$$P(L_R) = 1/(1 + e^{-z}) \quad (5)$$

where $z = \beta\ (V(L_R) - V(L_L))$ and the free parameter $\beta$ controls the degree of stochasticity observed in the choices. We fitted the data by maximizing the log-likelihood and choosing the best structural model to describe the monkeys' behavior using AIC[60].

$$AIC_{Model} = -2L + 2k \quad (7)$$

where $L$ is the maximum log-likelihood of the model and $k$ is the number of free parameters.

In each fitted model, whether $\alpha$, $\delta$, and $\gamma$ were significantly different from 0 was determined using a one-sample $t$-test at $P < 0.05$. Whether $\alpha$, $\delta$, and $\gamma$ were significantly different from one was also determined using a one-sample $t$-test at $P < 0.05$.

## Neural analysis
Peristimulus time histograms were drawn for each single-neuron activity, aligned at the onset of a visual cue. The average activity curves were smoothed using a 50 ms Gaussian kernel (σ = 50 ms). Basic firing properties, such as peak firing rates, peak latency, and duration of peak

activity (half-peak width), were compared among the four brain regions using parametric or nonparametric tests, with a statistical significance level of $P < 0.05$. Baseline firing rates 1 s before the appearance of central fixation targets were also compared with a statistical significance level of $P < 0.05$. These basic firing properties have been described by Yamada et al. (2021).

We analyzed neural activity during a 2.5 s period during pie chart stimulus presentation in the single-cue task. The firing rates of each neuron during the 1-s time window were estimated every 0.5 s after the onset of the cue stimuli. A Gaussian kernel was not used.

**Pre-screening neural activity for economic model fits.** To determine which neurons were sensitive to the probability and magnitude cued by a lottery, without assuming any specific model, the neural discharge rates ($F$) were regressed on a linear combination of a constant and the probability and magnitude of rewards:

$$F = b_0 + b_p p + b_m m \qquad (8)$$

where $p$ and $m$ are the probability and magnitude of rewards indicated by the pie chart, respectively. $b_0$ is the intercept. If $b_p$ and $b_m$ were not 0 at $P < 0.05$, the discharge rates were regarded as significantly modulated by that variable.

Based on linear regression, two types of neural modulations were identified: the "P + M + " type with a significant $b_p$ and a significant $b_m$ both having a positive sign (i.e., positive $b_p$ and positive $b_m$) and the "P-M-" type with a significant $b_p$ and a significant $b_m$ both having a negative sign (i.e., negative $b_p$ and negative $b_m$). Both types of neuronal signals represent the economic decision statistics described in the next section.

**Neural economic models.** We fitted the four neural models of the subjective valuation of lottery $L(p,m)$ to the activity of the preselected neurons that were sensitive to the information of probability and magnitude of rewards. The unified formula for all models is $R = g\,w(p)\,u(m) + b$, where the output of model $R$ represents the firing rates as a function of $V(p,m) = w(p)\,u(m)$, which is the subjective expected value function (SEV) of a lottery that reflects the lottery valuation of the neuron. Note that for the neural representation of $V(p,m)$, we call this value function different from the behavioral measures, the expected subjective utility. In all models, $b$ (baseline firing rate), $g$ (magnitude of the neural response), $\alpha$ (utility curvature), $\gamma$, and $\delta$ (probability weighting) were free parameters.

1. *Expected value model* (EV).

$$R = g\,p\,m + b \qquad (9)$$

2. *Expected utility model* (EU).

$$R = g\,p\,m^\alpha + b \qquad (10)$$

3. *Prospect theory model with one-parameter Prelec* (PT1).

$$R = g\,exp(-(-log\,p)^\gamma)m^\alpha + b \qquad (11)$$

4. *Prospect theory model with two-parameter Prelec* (PT2).

$$R = g\,exp(-\delta(-log\,p)^\gamma)m^\alpha + b \qquad (12)$$

To identify the structural models that best describe the activity of neurons in each brain region, we fitted each model to the P + M + and P-M- type activity of each neuron on a trial-by-trial basis. We estimated the combination of best-fit parameters using the R statistical software package the nls() function with random initial values (repeated 100 times) to find a set of parameters that minimized nonlinear least squared values.

For each of the four brain regions, the best-fit model showing the minimal AIC was selected by comparing the AIC values among the models. If the differences in AIC values against the three other models were significantly different from 0 in the one-sample $t$-test at $P < 0.05$, the model was defined as the best model. For a visual presentation, we plotted AIC differences in comparison to the EV model as the baseline model in the economics literature.

**Construction of the neural prospect theory model.** The estimated parameters in the best-fit model of neuronal activity were classified using PCA followed by the k-means clustering algorithm. PCA was applied once to all parameters estimated in the best-fit model PT2, that is, $b$, $g$, $\alpha$, $\delta$, and $\gamma$ in the DS, VS, and cOFC. The k-means algorithm was used to classify the five types of neural responses according to the PC1 to PC4 scores, as the first four PCs explained more than 90% of the variance. Following the classification, we defined each type of cluster with the mean of each estimated parameter as the five clusters were observed in each of the DS, VS, and cOFC neural populations.

**Evaluation of neural model performance using a network model for simulations.** We constructed a simple layered network model for simulations[32,33] with a minimal number of assumptions. We simply reconstructed a neural prospect theory model from the clusters above by accumulating each response $R$ of the five clusters. The first layer was composed of the five neural clusters (C1-C5: $R_1$-$R_5$) reflecting $V(p,m)$. To accumulate these $V(p,m)$ signals in the second layer, for clusters 1, 3, and 5, we linearly summed response $R$ in each of the 100 lotteries' conditions, while for clusters 2 and 4, which were mostly composed of P-M- types, we inversed their $R_j$ by subtraction ($R = R_1 - R_2 + R_3 - R_4 + R_5$). This is because signals of the P-M- types were negatively correlated with the $V(p,m)$. Then, this accumulated signal was filtered by a rectified linear unit (ReLU) function to remove negative values since it mimics the firing rate (i.e., $SEV = ReLU(R)$). We allocated them to the left and right target options to perform a simulation based on the difference in these integrated responses of neural clusters. For the third layer, we then simulated the percentage of right choices ($P\_Right$) for lottery pairs represented as four times all 10,000 combinations of two lotteries $L(p,m)$ using the logistic function

$$P\_Right = 1/(1 + e^{-z}) \qquad (13)$$

where $z = \beta\,(SEV(L_R) - SEV(L_L))$ and $\beta$ is assumed to be 1, i.e., no beta term. These simulated choice data composed of 40,000 choice trials were visualized and evaluated by applying the best-fit model to estimate the preference parameters $\alpha$, $\delta$, and $\gamma$ in $u(m) = m^\alpha$ and $w(p) = exp(-\delta\,(-log\,p)^\gamma)$, as well as $\beta$ in the choice function, similar to the model fit to the actual behavior of the monkeys. Thus, this simulation simply examined how the $V(p,m)$ neural signals distributed in the reward circuitry when monkey perceived probability and magnitude can reconstruct internal subjective valuation of risky prospects for economic choices.

To evaluate the statistical significance of the estimated internal preference parameters in the simulated choice data, we repeated the simulation to estimate the parameters $\alpha$, $\delta$, $\gamma$, and $\beta$ as above. We ran 1000 simulations to estimate the parameters and constructed the distribution of each parameter. The mean and standard deviation of each parameter were estimated. We then examined whether $\alpha$, $\delta$, and $\gamma$ were significantly different from 1 using each of these constructed distributions at $P < 0.05$. We also examined whether $\alpha$, $\delta$, and $\gamma$ were significantly different from those values estimated in each monkey's behavior at $P < 0.05$.

**Reporting summary**
Further information on research design is available in the Nature Research Reporting Summary linked to this article.

## Data availability

Data for the estimated parameter of the neural economic model is provided in the Supplementary Data file (Supplementary Data 1). See Supplementary information file for more details. Source data are provided with this paper.

## Code availability

Analysis code of the attached data and simulation code are provided in the Supplementary Code files (Supplementary Code 1 and Supplementary Code 2). See the Supplementary information file for more details.

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

## Acknowledgements

The authors would like to thank Takashi Kawai, Ryo Tajiri, Yoshiko Yabana, and Yuki Suwa for their technical assistance, and Jun Kunimatsu and Masafumi Nejime for their valuable comments. Monkey FU was provided by the NBRP "Japanese Monkeys" through the National Bio Resource Project of the MEXT, Japan. Funding: This research was supported by JSPS KAKENHI (Grant Numbers JP:15H05374, 19H05007, and 21H02797), Takeda Science Foundation, Narishige Neuroscience Research Foundation, Moonshot R&D JPMJMS2294 (H.Y.), JSPS KAKENHI 19K12165 (Y.T.), and ARC DP190100489 (A.T.).

## Author contributions

H.Y. designed the study. H.Y. and Y.I. conducted experiments. M.M. conducted part of the experiment. H.Y. and A.T. developed the analytic tools. H.Y. and Y.T. conceptualized the simulation tools. H.Y. and A.T. analyzed the data. H.Y., Y.T., and A.T. evaluated the result. H.Y. wrote the first draft of this manuscript. H.Y. and A.T. prepared the manuscript. Y.T. wrote a part of the manuscript. All authors have edited and approved the final manuscript.

## Competing interests

The authors declare no competing interests.
