## [Peer Review File · Nature Communications]

A neuronal prospect theory model in the brain reward circuitryREVIEWER COMMENTS

Reviewer #1 (Remarks to the Author):

This is a potentially interesting article in the field of neuroeconomics that addresses an important issue – whether the responses of individual neurons in monkeys passively viewing a lottery obey a prominent theory of choice, prospect theory, that has been previously validated exclusively in human neuroimaging studies. However, given the poor spatial and temporal resolution of functional imaging, it has been unclear whether single-unit neural activity is correlated with subjective valuation of risky gambles. They recorded in 4 areas of the reward circuit, one at a time, including the central part of the orbitofrontal cortex (cOFC), medial part of the orbitofrontal cortex (mOFC), dorsal striatum (DS), and ventral striatum. The authors found individual neurons whose activity can be described using the prospect theory model as a multiplicative combination of subjective utility and subjective probability functions. They further built a simple network model using the distributed subjective valuation signals to reconstruct animal's risk preference and subjective probability perception.

Overall, this is a reasonably well-designed study that is technically sound, despite the absence of state of art methodology, and presents results which, although not totally unexpected, contribute to previous neuroeconomics studies. Although I am mildly positive, the major shortcoming of the present version of the manuscript is the lack of convincing evidence to support the general conclusion. This makes the paper difficult to evaluate in its present form.

Major issues:

1) The authors need to better motivate their study in the context of prospect theory. It is unclear why this specific theory was chosen, what makes it fundamental, and what were the findings of previous neuroimaging studies. It would also be useful to better link their results to those of previous imaging investigations and comment about potential discrepancies.

2) I have several problems with the way the data is presented. It is unclear how many cells were recorded in each monkey and whether prospect theory explains firing rates in each animal, or perhaps the data and model predictions were skewed towards a particular monkey (especially given the low percentage of cells modulated by the probability and magnitude of reward in the 4 areas they recorded from).

3) I found the surface maps in Figures 2, 3, and 4 very difficult to read and relate to the neural findings and the text. Can they present the data in a different format that makes it more interpretable?

4) There is not much information provided about the multi-layer model described in Figure 5. How was the model constructed and then used to generate outputs? There is no statistical measure provided regarding model simulations, so it is difficult to assess the model performance and its validity.

Reviewer #2 (Remarks to the Author):

This is an interesting, novel, systematic and theory-driven study on the neuronal processing of the two most fundamental economic decision variables, using the well-established framework of economic choice theory. The authors have studied the two main variables, subjective utility and subjectively weighted probability, to investigate neuronal responses in four main reward areas of the brain. The paper is underpowered in terms of presentation of original neuronal data but presents well the results of proper statistical analyses for explaining choice behavior and neuronal responses. The presentation of original neuronal data needs to be extended before I would recommend publication. The remainder of my comments can be more easily addressed.

Data presentation: the neuronal responses need to be properly documented:

- Fig. 1h shows some neuronal changes with changing reward magnitude and probability, but it is unclear how these changes relate to the stated 'three different levels of probability (0.1–0.3, 0.4–0.7, 0.8–1.0) and magnitude (0.1–0.3 mL, 0.4–0.7 mL, 0.8–1.0 mL) of rewards' (legend). As these data are the core empirical results, they need to be presented in the proper standard used for the past 60 years, i.e. as rasters below histograms (PSTHs), and separately for each probability (not grouped over all probabilities). Only then can we see how neuronal activity changes with separate increases of magnitude and probability or expected value (EV). This will likely require a separate figure.
- These data need to be presented separately for each of the four investigated brain structures (maybe not all neuron displays with rasters).
- Also, it is unclear which task event elicited each response. Panel i provides some evidence in this direction, but we really need a proper breakdown according to task event, and for each task event (not just time inside a trial). And we need this separately for each of the four brain structures. So, maybe one neuronal response for each task event for one brain structure in a main figure, and the same for the other three brain structures in the suppl material.
- Further, we need population averages for the results described above (separately for each monkey), which can be distributed between main paper and suppl material.
- The associated text descriptions in the Results section are quite cryptic (lines 188-190). These descriptions need to be expanded according to the added figure documentation.

The regression results on lines 179-180 are presented in an unconventional way. They likely derive from the regression defined on line 660 (Methods), but results from such regressions are usually stated as proportion of variance explained (R^2) that varies between 0 and 1. The current text states an r (not r^2) of >1 . Please use conventional presentation of regression data (r^2 , β =slope, and significance of slope from zero, but maybe not original t -values).

The statistical model fittings of nonlinear utility and nonlinear probability weighting functions to behavioral choices and, separately, to neuronal responses are very detailed, nice and convincing (Figs. 2-4). Thank you very much.

The results from the model shown in fig 5 are interpreted to 'explain the internal subjective valuations of monkeys'. To make this a valid statement about subjectivity, the authors would need to use reward magnitude, instead of nonlinear utility, and linear probability, instead of a nonlinear probability function, with the same model and demonstrate better correspondence to the neuronal data when using utility and subjective probability compared to using objective reward magnitude and linear probability. Unless these comparisons are done, the model with fig 5 does not support the argument of 'subjective valuations' and has doubtful value for the argument of subjective coding.

There are a number of scholarship / wording issues:

- verbal expression: line 63-64: 'subjective perception of the reward probability': there is a difference between subjectively weighted probability perception and subjective weighting of the influence of probability on reward value. Both are captured indiscriminately by the usually nonlinear function $w(p)$ in the equation on line 198. So far, evidence suggests that the weight of probability on expected utility, Prospect or R (line 198) is subjective ($w(p)$), but it is unknown whether weighted probability perception is the reason for the observed subjective function $w(p)$. Please adjust the writing to the evidence.

- verbal expression: line 82: by gambling behavior, do you mean risky choice or true gambling? The phrasing as it stands could be misleadingly interpreted as gambling disorder (which is not the issue here). Please adjust the writing.

- misquotations: lines 82-83: 'Recordings of single-neuron activity in monkeys during gambling behavior may offer substantial progress over existing neuroimaging studies 7, 10, 11, 12'. None of the cited studies investigated single-neuron activity in monkeys. Please correct.

- the parameter g in the equation on line 198 is the neuronal response slope. Calling it 'the magnitude of the neural responses' is incorrect, even though the slope (first derivative of magnitude) determines response magnitude.

- the authors seem to have overlooked an earlier report that described utility coding by dopamine neurons in much detail (without subjective probability weighting) (Stauffer et al. Current Biology 2014), which should be cited.

Point-by-point reply to the reviewer's comments

We greatly appreciate the critical comments from the two reviewers. We have carefully reviewed each comment and fully revised our manuscript accordingly. We believe that addressing the comments and advice from the two reviewers has improved the introduction, results, methods, and discussion sections in our manuscript, which made our conclusion more convincing, thereby strengthening our manuscript. Below, we have provided our point-by-point responses to reviewers' comments.

Manuscript NCOMMS-22-05245-T

"A neuronal prospect theory model in the brain reward circuitry"

Since a new figure was added in this revision, the figure numbers changed as follows.

Previous	Current
Figure 1a-f	Figure 1
Figure 1f-i	Figure 2 (new figure panels were added)
Figure 2	Figure 3
Figure 3	Figure 4
Figure 4	Figure 5
Figure 5	Figure 6
-----	Figure S1

Reviewers' comments:

REVIEWER COMMENTS

Reviewer #1 (Remarks to the Author):

This is a potentially interesting article in the field of neuroeconomics that addresses an important issue – whether the responses of individual neurons in monkeys passively viewing a lottery obey a prominent theory of choice, prospect theory, that has been previously validated exclusively in human neuroimaging studies. However, given the poor spatial and temporal resolution of functional imaging, it has been unclear whether single-unit neural activity is correlated with subjective valuation of risky gambles. They recorded in 4 areas of the reward circuit, one at a time, including the central part of the orbitofrontal cortex (cOFC), medial part

of the orbitofrontal cortex (mOFC), dorsal striatum (DS), and ventral striatum. The authors found individual neurons whose activity can be described using the prospect theory model as a multiplicative combination of subjective utility and subjective probability functions. They further built a simple network model using the distributed subjective valuation signals to reconstruct animal's risk preference and subjective probability perception.

Overall, this is a reasonably well-designed study that is technically sound, despite the absence of state of art methodology, and presents results which, although not totally unexpected, contribute to previous neuroeconomics studies. Although I am mildly positive, the major shortcoming of the present version of the manuscript is the lack of convincing evidence to support the general conclusion. This makes the paper difficult to evaluate in its present form.

Response: Thank you for the positive evaluation of our work. We have addressed the specific reviewer's comments to the best of our ability below.

Major issues:

1) The authors need to better motivate their study in the context of prospect theory. It is unclear why this specific theory was chosen, what makes it fundamental, and what were the findings of previous neuroimaging studies. It would also be useful to better link their results to those of previous imaging investigations and comment about potential discrepancies.

Response: We appreciate that the reviewer encouraged us to provide a stronger contextual background for our choice to model neural activity with prospect theory. Thus, we have substantially rewritten the first two paragraphs of the introduction, where we introduce prospect theory as the leading theory of choice, which has still not been validated in individual neuron activity. We also briefly describe findings of previous studies in the second half of the second paragraph in the introduction.

Introduction, Page 4-5, Lines 60-92

"Since its inception in the 70s, prospect theory ¹ remains one of the most influential descriptive theories of choice in science and social science. The theory proposes that people calculate subjective valuations of risky prospects by a multiplicative combination of **two quantities**: a value function that captures the **subjective value** of rewards (i.e., utility) and an inverse S-shaped probability weighting function (i.e.,

probability weight) that captures a person's subjective distortion of the reward probability when calculating expected utility. The addition of the probability weighting function in their descriptive model of choice under uncertainty allowed Kahneman and Tversky to capture systematic deviations from the expected utility theory, such as Allais Paradoxes ² and the fourfold pattern of risk attitudes ³. Prospect theory has been assessed in thousands of studies using behavioral data and is used to explain a plethora of behaviors. However, despite the significant progress in the nascent field of neuroeconomics toward an understanding of how the brain makes economic decisions ^{4, 5}, a fundamental question that remains unanswered is whether discharges from individual neurons actually follow the prospect theory model. Does neuronal activity represent the multiplicative combination of subjective value and probability weighting functions?

Human neuroimaging provides fundamental insights into how economic decision-making is processed by brain activity, especially in the reward circuitry across cortical and subcortical structures ⁶. This circuitry is thought to learn the values of rewards and the probability of receiving them through experience ^{7, 8} and it allows human decision-makers to compute subjective valuations of options. Early research in neuroeconomics established that in line with economic theory ⁹, the brain encodes a utility-like signal that guides choice ¹⁰. At the same time, to establish a biologically viable, unified framework explaining economic decision-making under uncertainty, neuroeconomists aimed to incorporate not only the reward magnitude but also probability into the framework and searched for evidence of inverse-S subjective reward probability weighting in human brain activity using neuroimaging techniques ^{11, 12, 13, 14, 15}. Focusing on the gain domain ^{13, 15}, previous studies found that the activity of brain regions in the reward circuitry correlates with individual subjective valuations as proposed by the prospect theory ^{13, 15, 16}. However, limitations in temporal and spatial resolutions in neuroimaging techniques have restricted our understanding of how the reward circuitry computes subjective valuations of economic decisions, and there have been almost no studies involving the prospect theory analysis of neural mechanisms in the last decade."

As the reviewer suggested to describe the potential discrepancies between previous studies and the present study, we compared them in detail to key human fMRI studies (Abler et al., 2006; Preuschoff et al., 2006; Tobler et al., 2007; Berns et

al., 2008; Hsu et al., 2009) in the first paragraph of the discussion.

Discussion, P15, Lines 343-348

“This is consistent with the idea suggested by Tobler et al. (2007)³⁴ that the striatum integrates the reward magnitude and probability via multiplication into an expected value signal. Previous human fMRI studies found a nonlinear response to probability in striatal regions^{12, 15} and the dorsolateral prefrontal cortex³⁴, while some studies concluded that the probability coding in the striatum is linear^{11, 13, 14}. The probability weighting that we estimated in behavior and recovered from neural activity is concave.”

2) I have several problems with the way the data is presented. It is unclear how many cells were recorded in each monkey and whether prospect theory explains firing rates in each animal, or perhaps the data and model predictions were skewed towards a particular monkey (especially given the low percentage of cells modulated by the probability and magnitude of reward in the 4 areas they recorded from).

Response: Thank you for raising these issues regarding data presentation. To improve the visibility of our data, we have presented our data corresponding to each of the reviewer’s comments below.

> how many cells were recorded for each monkey?

This is a critical issue for reliability of the data showing whether the difference exists in our data between the two monkeys. As the reviewer suggested, we added the description for the number of recorded neurons in each monkey.

Page 9, Lines 181-184

“from neurons in the DS (n = 194: monkey SUN, 98; monkey FU, 96), VS (n = 144: monkey SUN, 89; monkey FU, 55), cOFC (n = 190: monkey SUN, 98; monkey FU, 92), and mOFC (n = 158: monkey SUN, 64; monkey FU, 94) (Fig. 2a).”

> whether prospect theory explains firing rates in each animal, or perhaps the data and model predictions were skewed towards a particular monkey (especially given the low percentage of cells modulated by the probability and magnitude of reward in the 4 areas they recorded from).

The reviewer suggested to show whether prospect theory can explain the firing rate in *each animal* since our previous manuscript lacked data presentations in each monkey.

In our analysis, we first identified neurons that show the potential neuronal signature of $V(p,m)$ (i.e., decision statistics used in economic models) using linear regression to detect the neural modulations both with the probability and magnitude of rewards. This pre-selection approach gave us an advantage in the next step, in which we could fairly compare which of the series of economic models better explained the firing rate of these neurons, such as the expected values, expected utility, and prospect theory models. As the reviewer enquired above, we did not show the data for each monkey for this critical process. Thus, we checked whether the percentages of these pre-selected neurons were different between monkeys in each brain region and checked population activity of neurons in the four brain regions in each monkey.

As a result of our analysis for the proportion of the pre-selected neurons, we found no significant differences in the proportion of neurons modulated by the probability and magnitude in the cOFC and VS, which were the predominant brain regions signaling the $V(p,m)$ (VS: monkey SUN 30/89, monkey FU 15/52, $\chi^2 = 0.17$, $df = 1$, $P = 0.682$; cOFC: monkey SUN 33/98, monkey FU 26/92, $\chi^2 = 0.42$, $df = 1$, $P = 0.52$). In contrast, we found a significant difference in the proportion of neurons signaling the $V(p,m)$ in both the DS and mOFC (DS: monkey SUN 32/98, monkey FU 11/96, $\chi^2 = 11.43$, $df = 1$, $P < 0.001$; mOFC: monkey SUN 15/64; monkey FU 2/94, $\chi^2 = 16.26$, $df = 1$, $P < 0.001$). These differences were clearly important, and thus, we added the following description in the Results section.

Page 10, Lines 214-218

“with significant differences in DS and mOFC between monkeys (DS: monkey SUN 32/98, monkey FU 11/96, $X^2 = 11.43$, $df = 1$, $P < 0.001$; VS: monkey SUN 30/89, monkey FU 15/52, $X^2 = 0.17$, $df = 1$, $P = 0.682$; cOFC: monkey SUN 33/98, monkey FU 26/92, $X^2 = 0.42$, $df = 1$, $P = 0.52$; mOFC: monkey SUN 15/64, monkey FU 2/94, $X^2 = 16.26$, $df = 1$, $P < 0.001$).”

We also added the population activity in each of four brain regions in each monkey in Fig. S1 and its legend as follows.

“Fig. S1. Population activities encoding probability and magnitude of rewards in the four brain regions

a Population activities of the $P+M+$ type recorded in the DS in monkey SUN for 10 levels of probability and magnitude of rewards. P and M indicate the probability and magnitude of rewards, respectively. Firing rates were normalized by the maximum firing rates among the combination of 100 lotteries. A curvature plot of population activities for the 100 lotteries is shown on the right. Average smoothing was made between neighboring pixels; n indicates the number of activities detected among

the four analysis epochs. **b** similar to **a** but for monkey FU. **c-d** similar to **a-b** but for P-M- type. **e-h** similar to **a-d** but recorded in the DS. **i-l** similar to **a-d** but recorded in the cOFC. **m-o** similar to **a-c** but recorded in the mOFC. No neurons for the P-M- type were recorded in the mOFC in monkey FU.”

As seen in the detected activity signaling $V(p,m)$ (see n values in each panel of the Fig. S1), signals from monkey FU were fewer those from monkey SUN. In contrast, firing rates of each neural population as shown in the curvature plot (see Y axis values) did not show clear differences between monkeys (Fig. S1 in the Results section). We denoted these differences between monkeys in result and discussion sections as follows.

Result, Page 10, lines 208-210.

“Similarly, some neurons showed activity modulated by both the probability and magnitude of rewards with negative regression coefficients, representing a negative coding type (P-M- type, Fig. S1).”

We also improved the discussion about the percent differences of potential signature of $V(p,m)$ in the DS and mOFC as follows.

Page 16, Line 365-367

“Our results from the reward-related brain regions are in line with this view, except for the mOFC, where fewer encodings of probability and magnitude of rewards were observed (Fig. 2h), with a significant difference between monkeys.”

Page 17, Lines 375-378

“Indeed, stable neural population dynamics in the VS and cOFC were observed in contrast to the fluctuating signals in the DS population ²⁸ with a significant difference between monkeys in the present study, which may reflect some dynamic differences in distributed coding in each individual.”

Discussion, Page 16, lines 355-358.

“It is important to denote that influences of $V(p,m)$ signals on the simulation might be different between monkeys, since larger number of the detected $V(p,m)$ signals in monkey SUN must affect the clustering of $V(p,m)$ signals.”

The reviewer also pointed out that percentages of the pre-selected neurons modulated by probability and magnitude are not high in all of the four brain regions we recorded. This is because other types of neurons whose activity was modulated by either probability or magnitude of rewards exist. We think that this is also an important point for describing the basic property of neurons we recorded, and thus, we added the following description in the Results section.

Page 10, Lines 222-227

We also found that the activity of neurons modulated by either probability or magnitude (probability, 305/686 neurons; magnitude, 269/686 neurons; at least one of the four analysis epochs) and across the entire cue period (probability: 0~1 s 108/686, 0.5~1.5 s 133/686, 1~2 s 128/686, 1.5~2.5 s 146/686; magnitude: 0~1 s 115/686, 0.5~1.5 s 113/686, 1~2 s 108/686, 1.5~2.5 s 113/686). We did not further analyze this activity of neurons because our main focus was on the $V(p,m)$.

We appreciate the many careful comments from the reviewer.

3) I found the surface maps in Figures 2, 3, and 4 very difficult to read and relate to the neural findings and the text. Can they present the data in a different format that makes it more interpretable?

Response: The reviewer pointed out that the surface map presentation is difficult to understand. This point is also related to the first and second comments from the second reviewer, which suggest that we need to present our data in the proper standard used for the past 60 years, i.e., as rasters below histograms (PSTHs). The second reviewer also suggests that we present neural activity according to the probability and magnitude of rewards in a more visible manner. Thus, we improved our presentation by plotting the neural firing rates in terms of the 10 levels of probability and magnitude of rewards separately in the new Figure 2f. We hope that this data presentation format may allow readers to connect average responses to each of the probability and magnitude of rewards to the surface map.

We added the new Figure 2 and its legend as follows.

“Fig. 2. Neural coding of probability and magnitude of rewards in the four brain regions

a Illustration of neural recording areas based on coronal magnetic resonance images. **b** Example activity histogram of a DS neuron modulated by probability and magnitude of rewards with positive regression coefficients during the single-cue task (*P+M+* type). The activity aligned to the cue onset is represented for three different levels of probability (0.1–0.3, 0.4–0.7, and 0.8–1.0) and magnitude (0.1–0.3 mL, 0.4–0.7 mL, and 0.8–1.0 mL) of rewards. Gray hatched time windows indicate the 1-s time window used to estimate the neural firing rates shown in **f** and **g**. Raster grams are shown below. **c-e** similar to **b**, but for VS, cOFC, and mOFC neurons. **f** Plot of the neural firing rates during the 1-s time window in **b** for ten levels of probability and magnitude of rewards. The firings are normalized by the maximum firing rates. P and M indicate the probability and magnitude of rewards, respectively. **g** Color map of the neural firing rates during the 1-s time window in **b** for ten levels of probability and magnitude of rewards. Average smoothing was made between neighboring pixels. **h** Percentage of neurons modulated by probability and magnitude of rewards in the four core reward brain regions. Black indicates activity showing positive regression coefficients for probability and magnitude of rewards (*P+M+* type). Gray indicates activity showing the negative regression coefficients for probability and magnitude (*P-M-* type). **a**, **c**, and **d** have been previously published in Yamada et al., 2021.”

4) There is not much information provided about the multi-layer model described in Figure 5. How was the model constructed and then used to generate outputs? There is no statistical measure provided regarding model simulations, so it is difficult to assess the model performance and its validity.

Response: Thank you for the comment stating that not enough information was provided to the multi-layer model. We agree that our explanation was insufficient especially for the definition of each layer as well as the meaning of each definition. We fully revised our description of the model in the Methods section as follows.

Methods, Page 26-27, Lines 583-606

“We constructed a simple layered network model for simulations^{32,33} with a minimal number of assumptions. We simply reconstructed a neural prospect theory model from the clusters above by accumulating each response R of the five clusters. The first layer was composed of the five neural clusters (C1-C5: R_1 - R_5) reflecting $V(p,m)$. To accumulate these $V(p,m)$ signals in the second layer, for clusters 1, 3, and 5, we linearly summed response R in each of the 100 lotteries' conditions, while for clusters 2 and 4, which were mostly composed of P-M- types, we inverted their R_j by subtraction ($R=R_1-R_2+R_3-R_4+R_5$). This is because signals of the P-M- types were negatively correlated with the $V(p,m)$. Then, this accumulated signal was filtered by a rectified linear unit (ReLU) function to remove negative values since it mimics the firing rate (i.e., $SEV = \text{ReLU}(R)$). We allocated them to the left and right target options to perform a simulation based on the difference in these integrated responses of neural clusters. For the third layer, we then simulated the percentage of right choices (P_Right) for lottery pairs represented as four times all 10,000 combinations of two lotteries $L(p,m)$ using the logistic function

$$P_Right = 1 / (1 + e^{-z})$$

where $z = \beta (SEV(L_R) - SEV(L_L))$ and β is assumed to be 1, i.e., no beta term. These simulated choice data composed of 40,000 choice trials were visualized and evaluated by applying the best-fit model to estimate the preference parameters α , δ , and γ in $u(m) = m^\alpha$ and $w(p) = \exp(-\delta (-\log p)^\gamma)$, as well as β in the choice function, similar to the model fit to the actual behavior of the monkeys. Thus, this simulation simply examined how the $V(p,m)$ neural signals distributed in the reward circuitry

when monkey perceived probability and magnitude can reconstruct internal subjective valuation of risky prospects for economic choices.”

The reviewer also pointed out that no statistical measure was provided to the simulated data. We performed the statistical test, and added the statistical results and its methods as follows.

Methods 27, lines 607-614

“To evaluate the statistical significance of the estimated internal preference parameters in the simulated choice data, we repeated the simulation to estimate the parameters α , δ , γ , and β as above. We ran 1,000 simulations to estimate the parameters and constructed the distribution of each parameter. The mean and standard deviation of each parameter were estimated. We then examined whether α , δ , and γ were significantly different from 1 using each of these constructed distributions at $P < 0.05$. We also examined whether α , δ , and γ were significantly different from those values estimated in each monkey’s behavior at $P < 0.05$.”

Results P15, lines 325-329

“We repeated this simulation 1,000 times to construct the parameter distributions of the internal subjective valuation obeyed by the layered model. The mean and standard deviation of the estimated parameters were as follows: alpha, 0.49 ± 0.017 ; delta, 0.50 ± 0.018 ; gamma, 1.67 ± 0.014 . They were significantly different from 1 ($P < 0.001$ for all cases).”

We appreciate the reviewer’s helpful comments.

Reviewer #2 (Remarks to the Author):

This is an interesting, novel, systematic and theory-driven study on the neuronal processing of the two most fundamental economic decision variables, using the well-established framework of economic choice theory. The authors have studied the two main variables, subjective utility and subjectively weighted probability, to investigate neuronal responses in four main reward areas of the brain. The paper is underpowered in terms of presentation of original neuronal data but presents well the results of proper statistical analyses for explaining choice behavior and neuronal responses. The presentation of original neuronal data needs to be extended before I would recommend publication. The remainder of my comments can be more easily addressed.

Response: Thank you for the positive evaluation of our work. We have addressed the specific reviewer's comments to the best of our ability below.

Data presentation: the neuronal responses need to be properly documented:

- Fig. 1h shows some neuronal changes with changing reward magnitude and probability, but it is unclear how these changes relate to the stated 'three different levels of probability (0.1–0.3, 0.4–0.7, 0.8–1.0) and magnitude (0.1–0.3 mL, 0.4–0.7 mL, 0.8–1.0 mL) of rewards' (legend). As these data are the core empirical results, they need to be presented in the proper standard used for the past 60 years, i.e. as rasters below histograms (PSTHs), and separately for each probability (not grouped over all probabilities). Only then can we see how neuronal activity changes with separate increases of magnitude and probability or expected value (EV). This will likely require a separate figure.

Response: The reviewer suggested to present our data in a more visible manner according to the standard presentation methods. To differentiate between probability and magnitude of rewards, raster plots below Fig. 1h and PSTHs in each probability levels are suggested. According to this reviewer's suggestion, we added the following two figures.

1. We added the raster grams for each of the three levels of probability and magnitude of rewards for Fig. 2b.
2. We improved our presentation by plotting the neural firing rates in terms of the 10

levels of probability and magnitude of rewards separately in the new Figure 2f. We hope that this data presentation format will allow readers to connect average response to each of probability and magnitude of rewards to the surface map.

We added the new Figure 2 and its legend as follows.

“Fig. 2. Neural coding of probability and magnitude of rewards in the four brain regions

a Illustration of neural recording areas based on coronal magnetic resonance images. **b** Example activity histogram of a DS neuron modulated by probability and magnitude of rewards with positive regression coefficients during the single-cue task (*P+M+* type). The activity aligned to the cue onset is represented for three different levels of probability (0.1–0.3, 0.4–0.7, and 0.8–1.0) and magnitude (0.1–0.3 mL, 0.4–0.7 mL, and 0.8–1.0 mL) of rewards. Gray hatched time windows indicate the 1-s time window used to estimate the neural firing rates shown in **f** and **g**. **Raster grams** are shown below. **c-e** similar to **b**, but for VS, cOFC, and mOFC neurons. **f** Plot of the neural firing rates during the 1-s time window in **b** for ten levels of probability and magnitude of rewards. The firings are normalized by the maximum firing rates. P and M indicate the probability and magnitude of rewards, respectively. **g** Color map of the neural firing rates during the 1-s time window in **b** for ten levels of probability and magnitude of rewards. Average smoothing was made between neighboring pixels. **h** Percentage of neurons modulated by

probability and magnitude of rewards in the four core reward brain regions. Black indicates activity showing positive regression coefficients for probability and magnitude of rewards (P+M+ type). Gray indicates activity showing the negative regression coefficients for probability and magnitude (P-M- type). **a, c, and d** have been previously published in Yamada et al., 2021.”

- These data need to be presented separately for each of the four investigated brain structures (maybe not all neuron displays with rasters).

Response: As described above, we added the three example neurons recorded from the VS, cOFC, and mOFC in the new Figure 2**c-e** as above. We also added the description and statistic for these additional three neurons in the Results section as follows.

Results, P9-10, lines 198-204

“These types of neurons were also observed in the VS, cOFC, and mOFC (Fig. 2c-e, VS, P+M+ type, percent variance explained, 0.440; probability, regression coefficient, $\beta = 7.14$, $t = 7.31$, $P < 0.001$; magnitude, $\beta = 6.71$, $t = 6.81$, $P < 0.001$; cOFC, P+M+ type, percent variance explained, 0.509; probability, regression coefficient, $\beta = 8.55$, $t = 6.91$, $P < 0.001$; magnitude, $\beta = 11.07$, $t = 8.95$, $P < 0.001$; mOFC, P+M+ type, percent variance explained, 0.238; probability, regression coefficient, $\beta = 2.72$, $t = 3.95$, $P < 0.001$; magnitude, $\beta = 2.88$, $t = 4.15$, $P < 0.001$)”

- Also, it is unclear which task event elicited each response. Panel i provides some evidence in this direction, but we really need a proper breakdown according to task event, and for each task event (not just time inside a trial). And we need this separately for each of the four brain structures. So, maybe one neuronal response for each task event for one brain structure in a main figure, and the same for the other three brain structures in the suppl material.

Response: Thank you for the precise evaluation of our work. This is our mistake in that we analyzed only the neural activity during the cue period. This confusion must come from the fact that we did not explain that the cue period activity must reflect the prediction of the outcome and hence, probability or magnitude or subjective expected values. To clearly explain our aim and analysis, we added the following explanation to the Results section.

Results, P9, lines 187-193

“Because after the presentation of lottery cue, neurons in the four brain regions showed a firing rate increase with a variable time course (Fig. 1h in Yamada et al., 2020), we analyzed the neural firing rate through the cue period with four analysis epochs (see Methods). These neurons were identified by regressing neural activity on probability and magnitude of rewards, and the neurons included in our analysis were those that had either both positive or both negative regression coefficients (see Methods), which is the potential signature of $V(p,m)$ - the decision statistics in economic theory.”

We further improved this issue by adding the information of the center event as a “cue” in the current Figure 2.

We appreciate the reviewer’s careful reading of our manuscript.

- Further, we need population averages for the results described above (separately for each monkey), which can be distributed between main paper and suppl material.

Response: Thank you for the reviewer’s clear advice to improve the visibility of our manuscript. We added the population average histograms in each monkey in each brain regions as Supplementary Figure S1. We also added the description of the results. We think that Figure S1 supports the visualization of our data. We appreciate the reviewer’s comment that this increases reliability of our manuscript.

Results, P10, Lines 208-210

“Similarly, some neurons showed activity modulated by both the probability and magnitude of rewards with negative regression coefficients, representing a negative coding type (P-M- type, Fig. S1).”

Figure legend

“Fig. S1. Population activities encoding probability and magnitude of rewards in the four brain regions

a Population activities of the $P+M+$ type recorded in the DS in monkey SUN for 10 levels of probability and magnitude of rewards. P and M indicate the probability and magnitude of rewards, respectively. Firing rates were normalized by the maximum

firing rates among the combination of 100 lotteries. A curvature plot of population activities for the 100 lotteries is shown on the right. Average smoothing was made between neighboring pixels; n indicates the number of activities detected among the four analysis epochs. **a** similar to **a** but for monkey FU. **c-d** similar to **a-b** but for *P-M*-type. **e-h** similar to **a-d** but recorded in the DS. **i-l** similar to **a-d** but recorded in the cOFC. **m-o** similar to **a-c** but recorded in the mOFC. No neurons for the *P-M*-type were recorded in the mOFC in monkey FU.”

- The associated text descriptions in the Results section are quite cryptic (lines 188-190). These descriptions need to be expanded according to the added figure documentation.

Response: We appreciate the reviewer’s careful comments on our description in the previous manuscript. We have changed many descriptions in the Results section as shown below.

Results, P9, Lines 180-227

“Neural signals for subjective valuations were distributed in the reward circuitry

We recorded single-neuron activity during the single-cue task (Fig. 1c) from neurons in the DS ($n = 194$: monkey SUN, 98; monkey FU, 96), VS ($n = 144$: monkey SUN, 89; monkey FU, 55), cOFC ($n = 190$: monkey SUN, 98; monkey FU, 92), and mOFC ($n = 158$: monkey SUN, 64; monkey FU, 94) (Fig. 2a). These brain regions are known to be involved in decision-making. We first identified neurons whose activity represents the key reward statistics—probability and magnitude—that underlie the expected value, expected utility, and prospect theory. Because after the presentation of lottery cue, neurons in the four brain regions showed a firing rate increase with a variable time course (Fig. 1h in Yamada et al., 2020), we analyzed the neural firing rate through the cue period with four analysis epochs (see Methods). These neurons were identified by regressing neural activity on probability and magnitude of rewards, and the neurons included in our analysis were those that had either both positive or both negative regression coefficients (see Methods), which is the potential signature of $V(p,m)$ - the decision statistics in economic theory.

An example of single neuron activity during a 1-s time window after cue onset is shown in Fig. 2b. This DS neuron showed activity modulated by both the probability and magnitude of rewards with positive regression coefficients ($P+M+$

type, percent variance explained, 0.462, $n = 114$; probability, regression coefficient, $\beta = 13.51$, $t = 8.57$, $P < 0.001$; magnitude, $\beta = 12.27$, $t = 7.79$, $P < 0.001$). These types of neurons were also observed in the VS, cOFC, and mOFC (Fig. 2c-e, VS, P+M+ type, percent variance explained, 0.440, $n = 115$; probability, regression coefficient, $\beta = 7.14$, $t = 7.31$, $P < 0.001$; magnitude, $\beta = 6.71$, $t = 6.81$, $P < 0.001$; cOFC, P+M+ type, percent variance explained, 0.509, $n = 119$; probability, regression coefficient, $\beta = 8.55$, $t = 6.91$, $P < 0.001$; magnitude, $\beta = 11.07$, $t = 8.95$, $P < 0.001$; mOFC, P+M+ type, percent variance explained, 0.238, $n = 120$; probability, regression coefficient, $\beta = 2.72$, $t = 3.95$, $P < 0.001$; magnitude, $\beta = 2.88$, $t = 4.15$, $P < 0.001$). Neuronal firing rates increased as the reward probability increased and as the reward magnitude increased, representing a positive coding type (Fig. 2f). In a plot of neuronal activity for all combinations of probability and magnitude, a curvature of the neural firing rates was detected (Fig. 2g). Similarly, some neurons showed activity modulated by both the probability and magnitude of rewards with negative regression coefficients, representing a negative coding type (P-M- type, Fig. S1). In total, these types of activity were observed in 24% (164/686) of all recorded neurons in at least one of the four analysis epochs during the 2.5-s cue period. The proportions of these signals in each brain region were different (DS, 22%, 43/194, VS, 32%, 45/141, cOFC, 31%, 59/190, mOFC, 11%, 17/158, chi-square test, $X^2 = 25.59$, $df = 3$, $P < 0.001$) with significant differences in DS and mOFC between monkeys (DS: monkey SUN 32/98, monkey FU 11/96, $X^2 = 11.43$, $df = 1$, $P < 0.001$; VS: monkey SUN 30/89, monkey FU 15/52, $X^2 = 0.17$, $df = 1$, $P = 0.682$; cOFC: monkey SUN 33/98, monkey FU 26/92, $X^2 = 0.42$, $df = 1$, $P = 0.52$; mOFC: monkey SUN 15/64, monkey FU 2/94, $X^2 = 16.26$, $df = 1$, $P < 0.001$). These neurons were evident across the entire cue period (Fig. 2h), during which the monkeys perceived the probability and magnitude of rewards. Thus, cue period activity in the four core reward brain regions showed potential signature of $V(p,m)$, which is the core decision statistics in economic theory.

We also found that the activity of neurons modulated by either probability or magnitude (probability, 305/686 neurons; magnitude, 269/686 neurons; at least one of the four analysis epochs) and across the entire cue period (probability: 0~1 s 108/686, 0.5~1.5 s 133/686, 1~2 s 128/686, 1.5~2.5 s 146/686; magnitude: 0~1 s 115/686, 0.5~1.5 s 113/686, 1~2 s 108/686, 1.5~2.5 s 113/686). We did not further analyze this activity of neurons because our main focus was on the $V(p,m)$."

The regression results on lines 179-180 are presented in an unconventional way. They likely derive from the regression defined on line 660 (Methods), but results from such regressions are usually stated as proportion of variance explained (R^2) that varies between 0 and 1. The current text states an r (not r^2) of >1 . Please use conventional presentation of regression data (r^2 , β =slope, and significance of slope from zero, but maybe not original t -values).

Response: Thank you for the helpful comment on the standard way to show regression results. As the reviewer suggests, we added the description of percent variance explained, which was shown in the previous Figure 3b and the current Figure 4b. We used “ β ” instead of “ r ” to represent the regression coefficient.

Results, P10, lines 196-198

“(P+M+ type, percent variance explained, 0.462; probability, regression coefficient, $\beta = 13.51$, $t = 8.57$, $P < 0.001$; magnitude, $\beta = 12.27$, $t = 7.79$, $P < 0.001$)”.

We also added the percent variance explained when we added new Figures 2c-e and their description in the main text.

The statistical model fittings of nonlinear utility and nonlinear probability weighting functions to behavioral choices and, separately, to neuronal responses are very detailed, nice and convincing (Figs. 2-4). Thank you very much.

Response: Thank you for the positive evaluation of our work. We are delighted with this reviewer’s comment.

The results from the model shown in fig 5 are interpreted to 'explain the internal subjective valuations of monkeys'. To make this a valid statement about subjectivity, the authors would need to use reward magnitude, instead of nonlinear utility, and linear probability, instead of a nonlinear probability function, with the same model and demonstrate better correspondence to the neuronal data when using utility and subjective probability compared to using objective reward magnitude and linear probability. Unless these comparisons are done, the model with fig 5 does not support the argument of 'subjective valuations' and has doubtful value for the argument of subjective coding.

Response: The reviewer pointed out that if subjective neural signal clusters 1–5 explain the internal subjective valuation of monkeys shown in the previous Figure 5, this results should be compared to objective coding of reward magnitude and probability on the firing rate of neurons. Since the objective coding of probability and magnitude of rewards (i.e., EV model) did not better explain the neuronal firing rate compared to subjective expected value model (shown in the previous Figure 3e and the current Figure 4e), we were not very sure about how to make this comparison with the same five clusters' model. One way we may make this comparison precisely is by assuming that all five clusters have linear probability and magnitude of rewards on their firing rates, i.e., alpha, gamma, and delta parameters were assumed to be 1, while g and b parameters were same as those in the original values in each cluster. Under this condition, neural coding of probability and magnitude of rewards was objective, and we could use the same model. We made a simulation with a same number of trials to sample and estimated internal subjective valuation of utility and probability weight from these simulated data, with alpha = 1, gamma = 1, and delta = 1.

As shown below, internal subjective valuation for the neural clusters with linear utility and linear probability weight was linear for both utility and magnitude of rewards. Thus, we believe that internal subjective representation of valuation on the neural firing rates must reflect the subjective valuation of risky rewards.

However, we are not very sure whether this assumption for this simulation is appropriate. Thus, we do not wish to add this result to our manuscript.

The reviewer also pointed out the other important issue to demonstrate correspondence to the neuronal data when using objective reward magnitude and linear probability. We calculated the AIC difference between EV model and the objective reward magnitude and probability model. As shown in the figure to the right,

fit performance between these two models was not significantly different from 0 (one sample t-test, Mean difference, 0.33, $t = 0.600$, $df = 298$, $p\text{-value} = 0.549$). Thus, the objective probability and magnitude model is not better than the EV model.

Because we are not very sure about the issues the reviewer pointed out above, we are happy to test more issues if our reply is not suitable for this comment. We appreciate the reviewer's comment on our simulation results.

There are a number of scholarship / wording issues:

- verbal expression: line 63-64: 'subjective perception of the reward probability': there is a difference between subjectively weighted probability perception and subjective weighting of the influence of probability on reward value. Both are captured indiscriminately by the usually nonlinear function $w(p)$ in the equation on line 198. So far, evidence suggests that the weight of probability on expected utility, Prospect or R (line 198) is subjective ($w(p)$), but it is unknown whether weighted probability perception is the reason for the observed subjective function $w(p)$. Please adjust the writing to the evidence.

Response: Thank you for pointing out the conceptual difference between distortions in perception and subjective weighting because they do not need to be the same thing. We removed the references to distorted "perception" throughout the manuscript.

- verbal expression: line 82: by gambling behavior, do you mean risky choice or true gambling? The phrasing as it stands could be misleadingly interpreted as gambling disorder (which is not the issue here). Please adjust the writing.

Response: Thank you for your comment. We changed the phrase "gambling behavior" to "risky rewards". We did not choose the phrase "risky choice" because some important previous studies recorded neural activity when monkeys receive risky rewards without choices. We changed the description in the Introduction as follows.

Introduction, P5, Lines 93 to 94

“Recordings of single-neuron activity in monkeys while receiving risky rewards^{17, 18, 19, 20, 21} may offer substantial progress over existing neuroimaging studies^{11, 12, 13, 14.}”

- misquotations: lines 82-83: 'Recordings of single-neuron activity in monkeys during gambling behavior may offer substantial progress over existing neuroimaging studies 7, 10, 11, 12'. None of the cited studies investigated single-neuron activity in monkeys. Please correct.

Response: We appreciate the reviewer’s comment. We added the reference for primate neural recording as follows.

Introduction, P5, lines 93-94.

Recordings of single-neuron activity in monkeys while receiving risky rewards^{17, 18, 19, 20, 21} may offer substantial progress over existing neuroimaging studies^{11, 12, 13, 14.}

References

17. McCoy AN, Platt ML. Risk-sensitive neurons in macaque posterior cingulate cortex. *Nat Neurosci* 8, 1220-1227 (2005).
18. O'Neill M, Schultz W. Coding of reward risk by orbitofrontal neurons is mostly distinct from coding of reward value. *Neuron* 68, 789-800 (2010).
19. So NY, Stuphorn V. Supplementary eye field encodes option and action value for saccades with variable reward. *J Neurophysiol* 104, 2634-2653 (2010).
20. Yang YP, Li X, Stuphorn V. Primate anterior insular cortex represents economic decision variables proposed by prospect theory. *Nat Commun* 13, 717 (2022).
21. Seo H, Cai X, Donahue CH, Lee D. Neural correlates of strategic reasoning during competitive games. *Science* 346, 340-343 (2014).

- the parameter g in the equation on line 198 is the neuronal response slope. Calling it 'the magnitude of the neural responses' is incorrect, even though the slope (first derivative of magnitude) determines response magnitude.

Response: Thank you for the careful evaluation of our description. As the reviewer pointed out, g in the equation does not indicate magnitude of the neural response. We realized that the g determines how much the neural firing rate depends on the

$w(p) u(m)$. Thus, we corrected the description as follows.

Results, P11, Lines 239-240

“ g determines how strongly the magnitude of the neural responses depends on the $u(m)$ and $w(p)$.”

- the authors seem to have overlooked an earlier report that described utility coding by dopamine neurons in much detail (without subjective probability weighting) (Stauffer et al. Current Biology 2014), which should be cited.

Response: We appreciate the reviewer’s comment. We added the description in the introduction with the reference.

Introduction, P5, lines 94-95,

“Specifically, utility coding without probability weighting was tested on activity of dopamine neurons²².”

REVIEWERS' COMMENTS

Reviewer #1 (Remarks to the Author):

I have no further comments. The authors have satisfactorily addressed my concerns.

Reviewer #2 (Remarks to the Author):

Thank you for your revision, which is perfect except for 2 points:

1) In line 196 (all line numbers refer to the revised paper), the authors say 'percent of variance explained' (percent is uncommon to use) but they state a number below 1.0 (which is far too low for a percentage; this is probably the R2 for the overall regression, or partial R2 for single regressors in multiple regressions, and an R2 is indeed <1): so maybe say 'proportion of variance explained' or just 'R2 for variance explained'. The issue comes back in lines 199, 201, 203, 251, fig 4b with line 882/3, fig 5b with line 896.

2) Interpretation of the empirical data with the model: In lines 264-275, the authors state that in all brain structures studied, nonlinear utility and nonlinear probability weighting fitted the neuronal data better than the EV model (linear utility, linear probability weighting), except for OFC (line 274). This is an empirical result and thus not an issue. The interpretation at the end of that Results section is fine as stated: 'coding underlies subjective valuation under risk' (line 302). But I find the interpretation of the model results imprecise and potentially misleading as it stands: 'explain the internal subjective valuations of monkeys' (line 330-331)'.

To demonstrate that the model shows subjective as opposed to objective neuronal value coding, I had suggested in the first round to test whether an EV model (linear utility, linear probability weighting) would reconstruct the neuronal data less accurately than a Prospect Theory model (nonlinear utility, nonlinear probability weight). The authors did such an analysis that resulted in a figure that they decided not to include in the paper (with 3 components in blue, orange and yellow). I don't understand that analysis, nor do the authors want to include it in the paper. Then they 'calculated the AIC difference between EV model and the objective reward magnitude and probability model': the EV model is about objective reward magnitude and probability, so I don't really understand what difference they calculated.

So, altogether, the authors do not present better model performance for subjective value (nonlinear utility, nonlinear probability weight) as compared to objective value (linear utility, linear probability weighting). Therefore, the model result as it stands does not disambiguate between subjective coding and objective coding. To avoid overinterpretation, the conclusion from the model of 'explain the internal

subjective valuations of monkeys' (line 330-331) needs to be amended / qualified / modified here, as well as in the Abstract (line 51), Introduction (line 112) and Discussion (line 354) by saying, for example, that the model also explains objective value (linear utility, linear probability weighting). This change does not affect the empirical data and does not change the overall conclusion of the paper.

Point-by-point reply to the reviewer's comments

We greatly appreciate the further comments from the second reviewer. As the reviewer suggests, we improved our manuscript corresponding to all comments from the reviewer. Below, we have provided our point-by-point responses to reviewers' comments.

Manuscript NCOMMS-22-05245A

"A neuronal prospect theory model in the brain reward circuitry"

Reviewers' comments:

Reviewer #1 (Remarks to the Author):

I have no further comments. The authors have satisfactorily addressed my concerns.

Response: We gratefully appreciate the reviewer's many efforts and useful comments through this review process.

Reviewer #2 (Remarks to the Author):

Thank you for your revision, which is perfect except for 2 points:

Response: We appreciate the two comments from the reviewer that further improve our manuscript. We improved our manuscript corresponding to these comments.

1) In line 196 (all line numbers refer to the revised paper), the authors say 'percent of variance explained' (percent is uncommon to use) but they state a number below 1.0 (which is far too low for a percentage; this is probably the R2 for the overall regression, or partial R2 for single regressors in multiple regressions, and an R2 is indeed <1): so maybe say 'proportion of variance explained' or just 'R2 for variance explained'. The issue comes back in lines 199, 201, 203, 251, fig 4b with line 882/3, fig 5b with line 896.

Response: As the reviewer suggest, we replaced “percent variance explained” to “proportion of variance explained” in the main text and in Figures 4 and 5.

2) Interpretation of the empirical data with the model: In lines 264-275, the authors state that in all brain structures studied, nonlinear utility and nonlinear probability weighting fitted the neuronal data better than the EV model (linear utility, linear probability weighting), except for OFC (line 274). This is an empirical result and thus not an issue. The interpretation at the end of that Results section is fine as stated: ‘coding underlies subjective valuation under risk’ (line 302). But I find the interpretation of the model results imprecise and potentially misleading as it stands: ‘explain the internal subjective valuations of monkeys’ (line 330-331)’.

To demonstrate that the model shows subjective as opposed to objective neuronal value coding, I had suggested in the first round to test whether an EV model (linear utility, linear probability weighting) would reconstruct the neuronal data less accurately than a Prospect Theory model (nonlinear utility, nonlinear probability weight). The authors did such an analysis that resulted in a figure that they decided not to include in the paper (with 3 components in blue, orange and yellow). I don’t understand that analysis, nor do the authors want to include it in the paper. Then they ‘calculated the AIC difference between EV model and the objective reward magnitude and probability model’: the EV model is about objective reward magnitude and probability, so I don’t really understand what difference they calculated.

So, altogether, the authors do not present better model performance for subjective value (nonlinear utility, nonlinear probability weight) as compared to objective value (linear utility, linear probability weighting). Therefore, the model result as it stands does not disambiguate between subjective coding and objective coding. To avoid overinterpretation, the conclusion from the model of ‘explain the internal subjective valuations of monkeys’ (line 330-331) needs to be amended / qualified / modified here, as well as in the Abstract (line 51), Introduction (line 112) and Discussion (line 354) by saying, for example, that the model also explains objective value (linear utility, linear probability weighting). This change does not affect the empirical data and does not change the overall conclusion of the paper.

Response: As the reviewer suggests, we modified our description to tone down at all the sentences the reviewer pointed out in the results (line 330-331) as well as the Abstract (line 51), Introduction (line 112) and Discussion (line 354). We described these changes at the last of this reply. Before explaining our corrections, we would like to explain our thought for this comment.

The main point the reviewer raised in this revision is that overinterpretation would occur because we did NOT show better model performance for subjective value (nonlinear utility, nonlinear probability weight) as compared to objective value (linear utility, linear probability weighting). He said above that “So, altogether, the authors do not present better model performance for subjective value (nonlinear utility, nonlinear probability weight) as compared to objective value (linear utility, linear probability weighting).” We agree to the reviewer’s point that without better model performance compared to the objective value coding (i.e., EV model), we cannot say that we found subjective value model coding (i.e., PT models). However, we have already showed the better model performance for subjective value coding (PT models) compared to objective value coding (EV model).

In this point of the comment, we think that some miscommunications between the reviewer and us may occur or we may not precisely follow his/her thought. This might be because we included other response when we reply to this comment. We explains these details in the next sections.

We have two objective value coding models. One is the model, which is represented as the “multiplicative combination” of linear utility and linear probability weighting. This is the model we have presented originally. The other is the model, which is represented as the “linear utility and linear probability weighting”, the reviewer described as above. We previously showed that both models show the lowest model performances among all subjective and objective coding models in the previous revision. We will explain again for each of the model performance.

In the first submitted manuscript, we have already showed better model performance for subjective value coding (PT models) as compared to objective value coding (EV model, which is multiplicative combination of linear utility and linear probability weighting), in Figure 4b for a single neuron and Figure 4e for population of neurons in each of four brain regions. The figure 4e showed the better model performance for subjective value model compared to the EV model defined as the multiplicative combination of linear utility and linear probability weighting.

When we performed the point-by-point reply in the first revision, we performed the analysis modeled by the “linear utility and linear probability weighting” raised by the reviewer (not the multiplicative EV model that we used in our original analysis). We compared the model performance for “linear utility and linear probability weighting” model relative to the multiplicative EV model, same as our analysis in our manuscript. We obtained the result that “linear utility and linear probability weighting” did not show better performance compared to the multiplicative EV

model. Thus, both multiplicative EV model and “linear utility and linear probability weighting” model showed the lowest performance to explain the neural activity, while the subjective value model (PT models) showed better performance.

In this point of the reviewer’s comment, we think that we may not precisely understand the point of the reviewer related to this model performance. Therefore, we amended our description to tone down at all the sentences the reviewer pointed out in the results (line 330-331), as well as in the Abstract (line 51), Introduction (line 112) and Discussion (line 354) as follows.

Abstract (line 51)

From

“A network model aggregating these signals **reliably** reconstructed the risk preferences and subjective probability weighting revealed by the animals’ choices.”

To

“A network model aggregating these signals reconstructed the risk preferences and subjective probability weighting revealed by the animals’ choices.”

Introduction (line 112)

From

“A simple network model that aggregates these subjective valuation signals, which are distributed through most parts of the reward circuitry, **successfully** reconstructed the monkey’s risk preference and subjective probability weighting estimated from the choices monkeys made in other situations.”

To

“A simple network model that aggregates these subjective valuation signals, which are distributed through most parts of the reward circuitry, reconstructed the monkey’s risk preference and subjective probability weighting estimated from the choices monkeys made in other situations.”

Results (line 330-331)

From

“Thus, we concluded that a distributed neural code that accumulates individual

neuronal signals **can** explain the internal subjective valuations of monkeys.”

To

“Thus, we concluded that a distributed neural code that accumulates individual neuronal signals **explains** the internal subjective valuations of monkeys.”

Discussion (line 354)

From

“suggesting that these subjective valuation signals in the reward circuitry **were** integrated into the brain to construct a decision output from risky perspectives.”

To

“suggesting that these subjective valuation signals in the reward circuitry **would be** integrated into the brain to construct a decision output from risky perspectives.”

We appreciate abundant efforts from the reviewer that makes our manuscript more precise and convincing.